Citation: *Molecular Systems Biology* 9:697
www.molecularsystemsbiology.com

*molecular* *systems* *biology*

# Temporal control of self-organized pattern formation without morphogen gradients in bacteria

Stephen Payne[1,5], Bochong Li[1,5], Yangxiaolu Cao[1], David Schaeffer[2], Marc D Ryser[2] and Lingchong You[1,3,4,*]

[1] Department of Biomedical Engineering, Duke University, Durham, NC, USA, [2] Department of Mathematics, Duke University, Durham, NC, USA, [3] Institute for Genome Sciences and Policy, Duke University, Durham, NC, USA and [4] Duke Center for Systems Biology, Durham, NC, USA
[5] These authors contributed equally to this work
* Corresponding author. Department of Biomedical Engineering, Duke University, CIEMAS 2355 101 Science Drive, Box 3382, Durham, NC 27708, USA.
Tel.: +1 919 660 8408; Fax: +1 919 668 0795; E-mail: you@duke.edu

**Diverse mechanisms have been proposed to explain biological pattern formation. Regardless of their specific molecular interactions, the majority of these mechanisms require morphogen gradients as the spatial cue, which are either predefined or generated as a part of the patterning process. However, using *Escherichia coli* programmed by a synthetic gene circuit, we demonstrate here the generation of robust, self-organized ring patterns of gene expression in the absence of an apparent morphogen gradient. Instead of being a spatial cue, the morphogen serves as a timing cue to trigger the formation and maintenance of the ring patterns. The timing mechanism enables the system to sense the domain size of the environment and generate patterns that scale accordingly. Our work defines a novel mechanism of pattern formation that has implications for understanding natural developmental processes.**

*Molecular Systems Biology* **9**:697; published online 8 October 2013; doi:10.1038/msb.2013.55
*Subject Categories:* synthetic biology
*Keywords:* morphogen; pattern formation; synthetic biology; systems biology; temporal control

## Introduction

A major challenge in biology is to better understand the mechanisms driving pattern formation in diverse processes, including slime mold aggregation (Keller and Segel, 1970; Goldbeter, 2006), stripe formation (Painter *et al*, 1999), and limb bud development (Verheyden and Sun, 2008). To date, primarily two types of mechanisms have been invoked to explain biological pattern formation. Both types of mechanisms rely on morphogens, or signaling molecules produced by a local source that activate a specific and distinct cellular response in a concentration-dependent manner (Borello and Pierani, 2010). The first relies on the notion of self-organization. A classic example is the Turing mechanism (Turing, 1952; Meinhardt and Gierer, 1974; Sick *et al*, 2006). By using interlocking positive- and negative-feedback loops, the mechanism can generate self-organized spatial patterns of two diffusible morphogens, which can then control patterns of downstream processes. The other, widely invoked mechanism is the French flag model (Wolpert, 1969; Casal *et al*, 2002). The essence of this mechanism is a pre-defined morphogen gradient that is interpreted by downstream genes, where different genes are activated at different ranges of morphogen concentration. The common thread connecting these two types of mechanisms is the requirement for morphogen gradients. Indeed, the generation and interpretation of morphogen gradients have been the focus of most studies on

biological pattern formation. However, using *E. coli* programmed by a synthetic gene circuit, we demonstrate here the formation of self-organized patterns without an apparent morphogen gradient. These patterns are self-organized in that they are not generated by pre-defined spatial cues.

Our circuit (Figure 1A; Supplementary Figure S1) consists of a mutant T7 RNA polymerase (T7 RNAP) (Tan *et al*, 2009) activating its own expression via a T7 promoter carrying a *lac* operator. T7 RNAP also activates expression of LuxR and LuxI. LuxI mediates synthesis of acyl-homoserine lactone (AHL), which can diffuse across the cell wall. When enough AHL accumulates in cell culture, intracellular AHL binds to and activates LuxR, which induces expression of T7 lysozyme. Lysozyme can inhibit T7 RNAP by forming a complex with it and preventing it from binding its cognate promoter (Supplementary Figure S2). To report the circuit dynamics, a cyan fluorescent protein (CFP) is co-expressed with T7 RNAP, and an mCherry protein is co-expressed with T7 lysozyme. The circuit can thus be divided into two modules: an activation module consisting of the T7 RNAP positive-feedback loop and an inhibition module consisting of quorum sensing-mediated lysozyme expression. Its logic resembles that of the classical Turing mechanism (Turing, 1952; Meinhardt and Gierer, 1974): activation is local since T7 RNAP is confined in the cells, whereas inhibition is global due to fast diffusion of AHL.

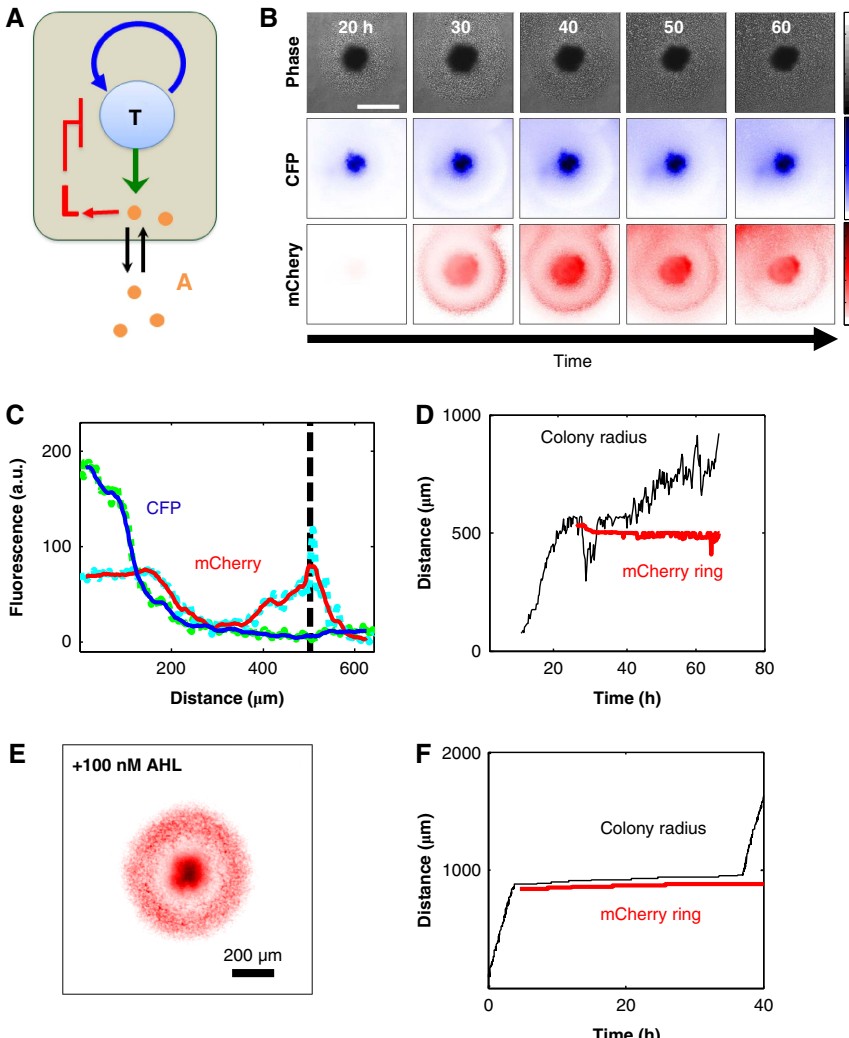

**Figure 1** Self-organized pattern formation in engineered bacteria. (**A**) Circuit logic. Our circuit consists of an activator T7 RNAP (T) activating itself and a diffusible signal, AHL (A). AHL can lead to repression of the activator by inducing T7 lysozyme (L). To monitor circuit dynamics, a CFP is co-expressed with T7 RNAP, and an mCherry is co-expressed with lysozyme (see Supplementary Figure S1 for further details). (**B**) The engineered bacteria developed a self-organized ring pattern. Images of a 1.2 mm × 1.2 mm field after 20, 30, 40, 50, and 60 h of incubation (as labeled). The microcolony was imaged using a Leica DM16000B fluorescence microscope with a mercury excitation lamp at 5X objective in the phase (first row), CFP (second row), and RFP (third row) channels. For the CFP and RFP images, the color scheme is defined by the darkest blue and darkest red representing saturation in the CFP and RFP channels, respectively, and white representing background levels. The phase images are raw images; the white scale bar on the 20-h phase image indicates a length scale of 500 μm. The scale bars to the right of each row represent the intensity scales for each image in its respective row, where the top indicates saturating intensity and the bottom indicates background intensity. (**C**) CFP (green dots) and mCherry (cyan dots) at the 30th hour at varying radial distance from the center. The solid blue and red lines are the running averages of the CFP and mCherry intensities, respectively. The black dashed line indicates the radial distance at which the running average of mCherry intensity is maximal outside of the core. This distance is defined as the mCherry ring radius plotted versus time in (**D**). Intensity values were calculated as the average intensity values across all angles at fixed radii about the microcolony core center. Each of these intensity values had background signal subtracted. This processing was carried out using a custom MATLAB algorithm. (**D**) mCherry ring radius (red line) and colony radius (black line) over time. The mCherry ring radius was calculated as described in (C). The colony radius was calculated as the distance from the center of the microcolony core to the microcolony edge averaged across angles spanning π/6 to π/4. Both computations were performed using a custom MATLAB algorithm. (**E**) mCherry image in the presence of 100 nM AHL. An mCherry bullseye pattern, albeit smaller pattern, still occurs after initial exogenous addition of 100 nM AHL. These data suggest that an AHL morphogen gradient is not necessary to obtain the mCherry bullseye pattern. The image is prepared as described in (**B**) row 3. (**F**) mCherry ring radius (red line) and colony radius (black line) over time. The base parameter set for the 1D simulation is listed in Supplementary Table S1. See Materials and methods for details. The y axis is distance from Δ = 0. Processing of the simulated data was done in the same way as for the experimental data in (**D**).

However, our circuit differs from the classical Turing mechanism in two critical aspects. First, in contrast to the Turing mechanism, which utilizes two diffusible morphogens, our circuit only contains a single morphogen: the AHL molecule. T7 RNAP transport is cell mediated and thus not driven by Fickian diffusion, which is assumed to take place in most reaction-diffusion models describing pattern formation. While underappreciated in theoretical studies, this mechanism of activator transport is likely common in natural developmental processes, such as stripe formation and limb bud outgrowth (Painter et al, 1999; Pfeiffer et al, 2000; Ibanes et al, 2006; Chisholm et al, 2010). Second, circuit activation induces

a metabolic burden on the growth of its host cell. This metabolic burden is critical for enhancing the non-linearity in the system (Tan *et al*, 2009) and exhibiting microcolony size control, which in turn facilitate robust pattern formation. Therefore, our circuit defines a tight coupling between intracellular gene expression, signal diffusion, and modulation of cell growth and motility. This coupling is often neglected in variants of Turing models invoked to explain self-organized pattern formation; yet, it is likely critical for diverse natural developmental processes, ranging from limb bud outgrowth (Verheyden and Sun, 2008) to somitogenesis (Cooke and Zeeman, 1976) to tissue stratification (Chou *et al*, 2010).

## Results

To explore the spatiotemporal dynamics of our system experimentally, we used a multi-well device (see Supplementary Figure S3) to culture microcolonies initiated from individual *E. coli* cells programmed by our gene circuit. Briefly, we added serially diluted overnight culture to molten soft agar containing growth medium supplemented with the appropriate antibiotics and 1000 μM isopropyl-1-thio-β-D-galactopyranoside (IPTG). Five-microliter droplets of the agar containing about 1–4 *E. coli* cells were then placed into wells of the multi-well device. We then observed the growth and gene expression of the expanding microcolonies in each well at 30°C.

Figure 1B shows growth and gene expression dynamics in a representative microcolony, which gives rise to a self-organized pattern with a length scale of ∼500 μm (also see Supplementary Movies 1 and 2). The pattern consisted of a CFP core and an mCherry ring (Figures 1B and C), whose formation appeared to be tightly coupled with colony expansion. Before 30 h of incubation, CFP slowly accumulated in the microcolony interior before mCherry underwent a switch-like transition from a low state to a high state in a ring pattern in <5 h. Initiation of the mCherry ring was concurrent with a pause in colony expansion (Figure 1D; Supplementary Movie 1). Though colony expansion resumed after ∼20 h, the ring size was maintained (Figure 1D; Supplementary Movie 2) for ∼35 h (until the end of the experiment). During this time, the intensity of the ring gradually decreased (Figure 1D; Supplementary Movie 2), suggesting that formation of the ring resulted from transient circuit dynamics. As indicated by additional results acquired under the same conditions, the dominant features of the self-organized patterns were robust and reproducible (Supplementary Figure S4). Similar patterns were also generated using a different strain (MG1655) grown under a different experimental condition (Supplementary Figure S5). These results again reinforce that our circuit is robust in that its characteristic behavior is persistent. Furthermore, this characteristic behavior is dependent on AHL signaling since the original cell strain (MC4100Z1) containing the full synthetic circuit with LuxR or LuxI removed did not generate a detectable and clear mCherry ring pattern (Supplementary Figure S6).

These results were counterintuitive considering the kinetic properties of the 'morphogen' AHL. It is a small molecule with an estimated diffusivity of 0.4 cm$^2$/h (Song *et al*, 2009). On the

time scale of our experiments (∼50 h), the diffusion length scale is ∼9 cm, which is about 2 orders of magnitude greater than that of our observed patterns. However, at the very early stage, AHL may form a transient gradient that could be critical for forming the pattern. To test this notion, we cultured cells in the presence of an initial concentration of AHL (100 nM). This concentration is well above the published estimate of the half-activation threshold of AHL (20 nM) (Collins *et al*, 2006) and is likely enough to saturate LuxR. Importantly, this concentration of AHL is well beyond what can be produced by the microcolony at the early stage. As such, introducing the AHL at this high concentration exogenously eliminates the possibility of a significant AHL gradient early in the experiment. Even under this condition, the microcolony was able to generate a similar (albeit smaller) pattern, confirming the ability of the system to generate patterns without requiring an AHL gradient (Figure 1E).

To help resolve this paradox, we developed an agent-based model (Ferreira *et al*, 2002; Mallet and De Pillis, 2006) (see Supplementary Information for details) to simulate the circuit-mediated spatiotemporal dynamics. Briefly, we modeled the cells as agents within which production and degradation of T7 RNAP, T7 lysozyme, and AHL take place. AHL degradation also takes place outside of the cell. We assume that cells undergo a random walk on a 1-dimensional (1D) spatial domain, where their growth and movement is sensitive to a metabolic burden induced by T7 RNAP (Tan *et al*, 2009) and T7 lysozyme (Studier, 1991). Due to very fast diffusion of AHL, we assume that its concentration is uniform across the entire spatial domain. In addition, we also assume that a cell's gene expression decreases with increasing distance from the microcolony edge via a variant of the Hill function. Through mathematical modeling, we have shown that spatial-dependent gene expression is essential in generating the patterns we observe. Several mechanisms can account for this spatial-dependent gene expression. These include contact inhibition (Morse *et al*, 2012) and mechanical stress (Jozefczuk *et al*, 2010) due to higher cell packing in the interior of the microcolony. Another possible contributing factor is that a cell's gene expression capacity increases with the availability of growth-limiting chemical substrates (Klumpp *et al*, 2009; Scott *et al*, 2010). Furthermore, gene expression capacity could increase with higher oxygen concentrations (Salmon *et al*, 2003) on the edge of the microcolony. Regardless of the underlying mechanisms, our assumption is consistent with our experimental observation (Supplementary Figure S7). Specifically, we observed the formation of the mCherry ring at the edge of growing microcolonies of MC4100Z1 cells expressing mCherry from a *ptet* promoter for several different time points. This result provides direct evidence for the spatial dependence of gene expression capacity.

Starting from 10 cells per spatial unit Δ, spanning Δ = 1–10, our model can reproduce the observed self-organized patterns (Figures 1F and 2A–C; Supplementary Figure S8) using biologically feasible parameters (see Supplementary Table S1). In particular, the simulation demonstrates several salient pattern features, including an initial confined T7 RNAP core, a T7 lysozyme ring with robust size over time, and the tight coupling between microcolony expansion and initiation and maintenance of the lysozyme ring (Figure 1F).

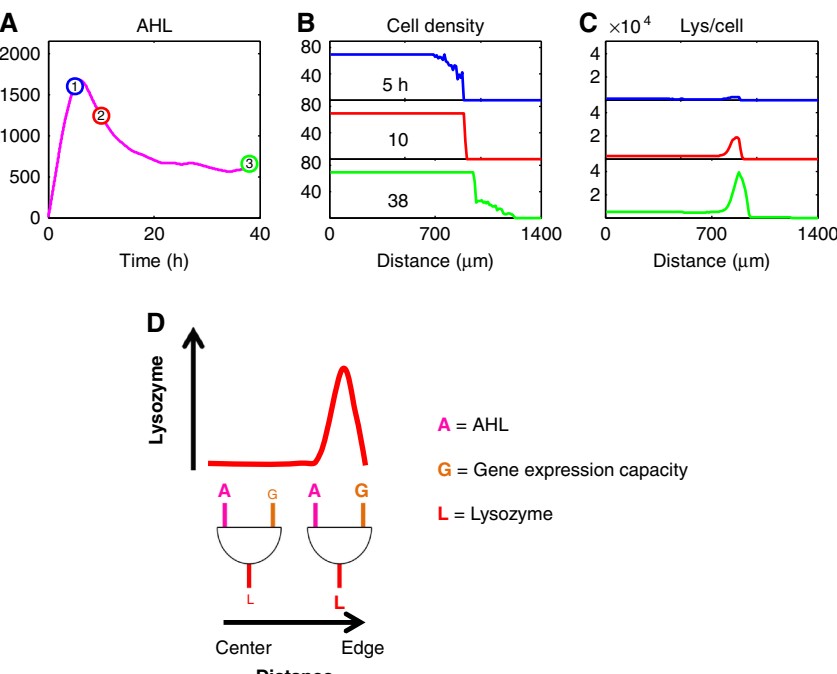

**Figure 2** Proposed mechanism for ring formation and maintenance. (**A**) AHL dynamics drive ring formation in the base case simulation. A single AHL temporal pulse gives rise to a single T7 lysozyme ring. Here, the *y* axis corresponds to the number of AHL molecules per spatial unit Δ. These dynamics are derived from the same simulation analyzed in Figure 1F with the same base parameter set listed in Supplementary Table S1. (**B**) Cell density dynamics for the base case simulation. Cell density is plotted as the number of cells per Δ for time points 1–3 of (**A**) (corresponding to 5, 10, and 38 h) from top to bottom. After AHL exceeds the ring-forming threshold, T7 lysozyme accumulates at the edge of the microcolony (**C**). Lysozyme induces a metabolic burden on cells at the edge of the microcolony, leading to a stunting of cell growth for several hours (top two panels). Once AHL decreases over time (since T7 lysozyme decreases AHL production), lysozyme's metabolic burden on the cells decreases at the very edge of expansion front, eventually leading to resumed cell growth (bottom panel). (**C**) Lysozyme dynamics for the base case simulation. T7 lysozyme is plotted as the number of T7 lysozyme molecules per cell for time points 1–3 of (**A**) (corresponding to 5, 10, and 38 h) from top to bottom. As AHL increases over time, T7 lysozyme accumulates at the edge of the spatial domain (top two panels). Eventually, lysozyme level decreases on the very edge of the microcolony (bottom panel), giving rise to microcolony growth resumption (B, bottom panel), while the position of the ring is maintained. (**D**) Lysozyme ring arises due to differential gene expression capacity throughout the microcolony. In the simulations, it is assumed that gene expression capacity increases with decreasing distance from the microcolony edge. Thus, T7 lysozyme production can be viewed as an AND gate, where both high AHL (A) and high gene expression capacity (G) are necessary to trigger lysozyme production. In this manner, at a time when A is uniformly high throughout the entire spatial domain, high A and low G give rise to low lysozyme levels toward the center of the microcolony. However, at the microcolony's edge, high A and high G give rise to high lysozyme levels (red line).

Importantly, the simulation confirms the notion that robust pattern formation does not require an AHL gradient. Instead, it underscores the critical interplay between cell growth and gene expression, which converts temporal information regarding the history of cell growth into spatial patterns (Figure 2; Supplementary Figure S8). Initially, AHL concentration is low due to diffusion-mediated dilution spanning the entire spatial domain. Thus, the early circuit dynamics are dominated by T7 RNAP-positive feedback.

The high T7 RNAP in the microcolony interior eventually gives rise to faster production of AHL, leading to a uniform elevation of AHL concentration throughout the entire spatial domain. Once AHL accumulates to a critical threshold level, it can activate T7 lysozyme production. This activation occurs on the colony edge, where the gene expression capacity of the cell is high, leading to an mCherry ring (Figures 2C and D). Toward the microcolony interior, however, low gene expression capacity results in low lysozyme production and mCherry expression. Thus, lysozyme synthesis can be viewed as the result of an AND logic gate: specifically, synthesis only occurs as the result of both high AHL and high gene expression capacity. This logic gate results in low mCherry in the microcolony interior and high mCherry at the microcolony edge at a uniformly high AHL concentration. These mCherry dynamics are confirmed by confocal imaging (Supplementary Figure S9), where we observe that the apparent mCherry core from fluorescence microscopy (Figure 1B) indeed is the result of another mCherry ring pattern occurring in the z-direction in the microcolony core. On the *x-y* plane where subsequent growth outside of the core is observed, a single mCherry ring forms on the microcolony edge as the model predicts. At the ring, expression of lysozyme causes a significant growth inhibition and a pause of colony expansion (Figures 1D, F and 2B; Supplementary Figure S8). Consequently, growth inhibition leads to a reduced dilution rate of lysozyme, reinforcing its accumulation. Thus, the interplay between expression of lysozyme and growth inhibition creates a local positive feedback, which facilitates the maintenance of the ring pattern.

Over time, AHL concentration gradually decreases due to reduced T7 RNAP strength (Figure 2A). As a result, lysozyme decays on the colony's very edge and thus lysozyme-induced metabolic burden is alleviated, leading to the resumption of cell growth and colony expansion. Now, AHL concentration is

too low to trigger expression of mCherry at the colony expansion front. As a result, the ring radius is maintained over time at its initial position in a manner that is decoupled from further colony expansion.

An open question in developmental biology is the exact mechanism by which pattern sizes are controlled by genetic and environmental factors. In our system, the critical determinant of pattern size is the timing of AHL accumulation, rather than its spatial gradient, which is negligible within the small dimensions of the microcolony. This timing can be modulated by changing the domain size or by simply adding exogenous AHL (Figure 3A). The negative-feedback loop mediated by the inhibition module has the potential to generate one or more AHL pulses. Indeed, the base-case circuit dynamics result in a single pulse of AHL, which triggers ring initiation when the AHL threshold is exceeded. The initial addition of exogenous AHL to the droplet is expected to decrease the time necessary to reach the AHL threshold and thus leads to earlier formation of a smaller ring (Figure 3A). In contrast, an increase in the domain size is expected to prolong the time necessary for AHL to reach the threshold due to greater spatial dilution, leading to later formation of a larger ring (Figure 3A).

We carried out further simulations and experiments to test these notions. Indeed, our simulations predict that the ring size decreases with increasing initial AHL concentration (Figure 3B; Supplementary Figure S10). This prediction was consistent with our experimental observation: the mCherry ring radius decreased from 655 to 282 μm as we increased the initial AHL concentration from 0 to 100 nM (Figure 3C). Similarly, our simulations predict that the ring size increases with domain size (Figure 3D; Supplementary Figures S11 and S12). Our experiments confirmed this prediction: mCherry ring radius increased from 572 to 1145 μm as we increased the droplet size from 5 to 15 μl (Figure 3E).

Finally, our simulations indicate that our system can generate a double-ring pattern when the AHL threshold necessary to initiate ring formation is exceeded twice during two discrete time intervals (Figures 4A and B). Multiple rings were obtained within a reasonable time frame most easily for high initial AHL concentrations that exceeded the ring-forming threshold from the start of colony formation (Figures 4A and B; Supplementary Figure S13). Again, this prediction was validated experimentally. A double-ring pattern was observed after 48 h of incubation for a single microcolony in a 5-μl droplet when supplied with an initial AHL concentration of 100 nM (Figures 4C and D). Note that in both the simulated and experimental cases for which a double ring occurs, the rings emerge sequentially with one ring first occurring at a smaller radius, followed by a second ring occurring at a larger radius (Figures 4B and D; Supplementary Figure S13). Here, the first ring corresponds to the initial time interval in which AHL exceeds the ring-forming threshold, and the second ring corresponds to the second time interval in which AHL exceeds the ring-forming threshold (Figures 4A and B; Supplementary Figures S13 and S14).

## Discussion

Our analysis has established a novel mechanism for forming self-organized patterns: instead of providing a spatial cue, the morphogen serves as a timing cue to initiate pattern formation. Thus, the pattern length scale is decoupled from that of the morphogen. These properties are fundamentally different from recent examples of synthetic pattern-forming circuits (Basu *et al*, 2005; Sohka *et al*, 2009; Liu *et al*, 2011), where the length scale of the pattern was determined by the length scale of the morphogen gradient. Our mechanism suggests a new way to view the role of morphogens involved in natural pattern-formation processes. Most past studies have focused on elucidating how morphogen gradients are generated and interpreted to realize precise patterns (Koch and Meinhardt, 1994; Maini *et al*, 2006; Rogers and Schier, 2011). However, less widely recognized potential pattern-formation mechanisms involving differential metabolism were proposed many decades ago (Parker, 1929; Child, 1941). Moreover, a recent study suggests that pattern formation in early sea urchin development does not require the participation of a morphogen (Smith *et al*, 2007). In addition, the role of mechanical forces has become increasingly appreciated in diverse pattern-forming biological systems (Gibson *et al*, 2011; Asally *et al*, 2012; Bosveld *et al*, 2012). Even when apparent morphogens are identified (Sick *et al*, 2006), their specific role in forming the resulting patterns has not been definitively established (Maini *et al*, 2006). Furthermore, a recent study has shown that a flat spatial distribution of the morphogen Bicoid can still give rise to well-defined head gap gene domains in *Drosophila* embryogenesis (Chen *et al*, 2012; Roth and Lynch, 2012). Taken together, these studies and our results suggest that the common view of a spatial morphogen gradient being the central driver of pattern formation may be biased and underestimate the contribution of the temporal morphogen dynamics, as well as that of other cellular processes, such as cell growth and movement.

In addition, the role of AHL in our system is analogous to that of secreted negative-feedback factors ('chalones') that are involved in tissue development (Lander *et al*, 2009). This concept, proposed five decades ago, explains organ size control as the result of slow accumulation of the chalone as cells proliferate. Once an organ reaches its target size, chalones accumulate to high enough concentrations to inhibit further organ growth (Bullough, 1962). While recent studies have identified candidate factors that fulfill the classical definition of the chalone (McPherron *et al*, 1997; Gamer *et al*, 2003; Lander *et al*, 2009), the specifics of how their local activity translates to organ size regulation is an open question (Gamer *et al*, 2003). By utilizing a synthetic system in which a diffusible signal (AHL) activates synthesis of the growth inhibitor (T7 lysozyme), we have eliminated many confounding factors influencing pattern-formation processes in natural systems. This strategy allowed us to observe that one possible method by which chalones regulate pattern size is through sensing the domain size by a fast-diffusing signal (Figures 3D and E). If so, our mechanism can provide an intuitive explanation for scale invariance observed in many developmental processes.

## Materials and methods

### Liquid medium

The 2xYT medium was made following the protocol in Sambrook and Russell (2001): 16 g tryptone (Difco Laboratories), 10 g yeast extract

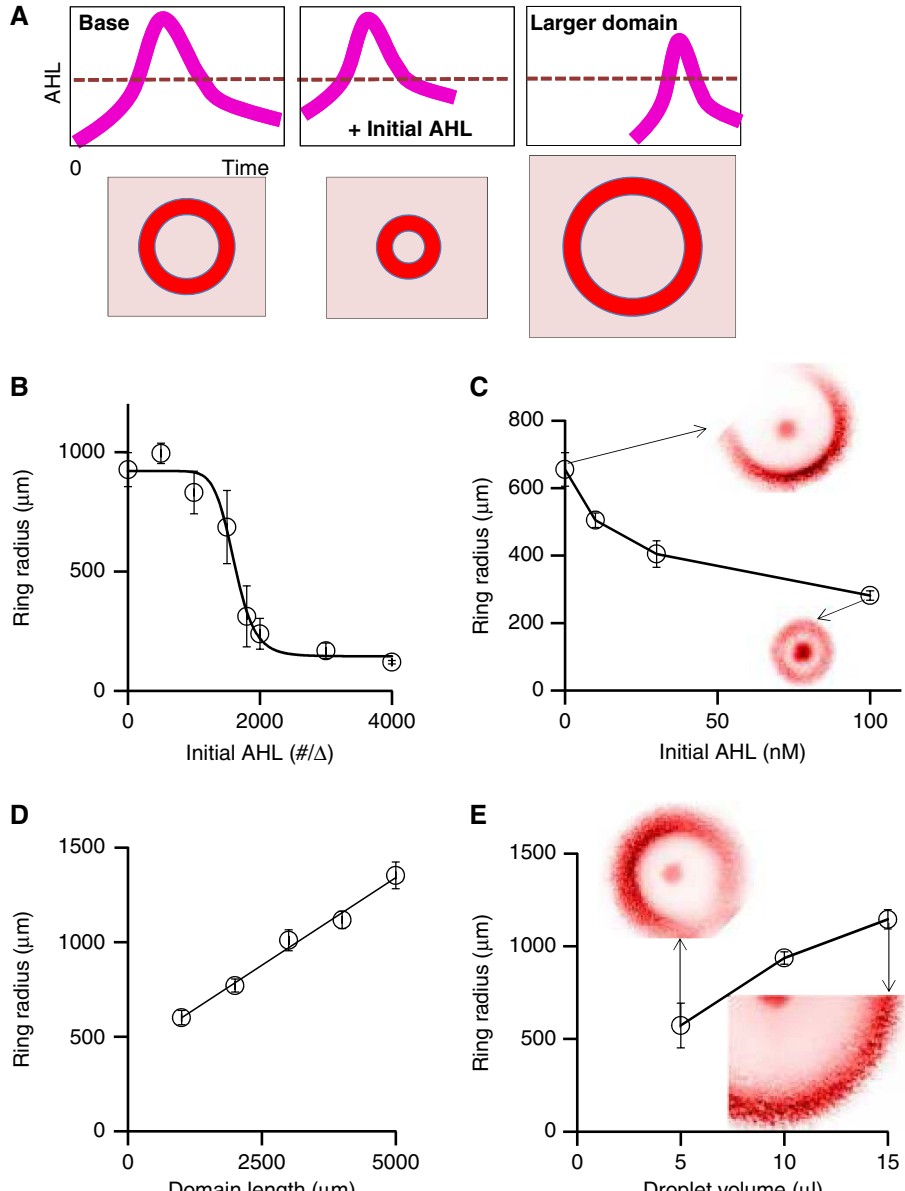

**Figure 3** Simulated and measured modulation of pattern formation by environmental factors. (**A**) Modulation of patterns by perturbing AHL temporal dynamics. The base case occurs when a single pulse of AHL exceeds a threshold necessary to trigger ring formation (left, e.g., as in Figures 1 and 2). Adding exogenous AHL allows AHL to exceed the threshold concentration faster, leading to the formation of a smaller ring (center). Increasing the domain size slows down AHL accumulation due to increased spatial dilution, leading to the formation of a larger ring (right). (**B**) Simulated dependence of ring radius on initial, exogenously added AHL concentration. Average mCherry ring radii obtained at 25 and 13.3 h for 1D simulations of microcolonies growing from initial AHL concentrations of 0–1200 molecules per $\Delta$ and 1500–4000 molecules per $\Delta$. For each replicate, the time chosen for data analysis corresponds to the time at which only the first mCherry ring radius has emerged. All of the mCherry ring radii were calculated in the same manner as described in Figure 1. The error bars represent standard error among 10 replicates. The black curve indicates a best-fit Hill function calculated using a custom MATLAB code. (**C**) Measured dependence of ring radius on initial, exogenously added AHL concentration. Average mCherry ring radii obtained for microcolonies growing from an initial AHL concentration of 0, 10, 30, and 100 nM AHL, respectively. These values were obtained from replicates at 24-h time points for 10, 30, and 100 nM AHL and from replicates at 36-h time points for 0 nM AHL (no mCherry rings emerged for this condition at the 24-h time point). All of the mCherry ring radii were calculated in the same manner as described in Figure 1. The error bars for 0, 10, 30 and 100 nM AHL represent standard error among 7, 10, 9, and 5 replicates, respectively. Cropped representative mCherry images to scale of microcolonies growing in the absence (top middle panel) or presence (bottom right panel) of an initial AHL concentration of 100 nM at 36-h and 24-h time points, respectively, are shown. The color scheme is defined as in Figure 1. (**D**) Simulated dependence of ring radius on the domain size. Average mCherry ring radii obtained at 25 h for 1D simulations of microcolonies growing in domain lengths spanning 100–500 $\Delta$ (1000–5000 μm). All of the mCherry ring radii were calculated in the same manner as described in Figure 1. The error bars represent standard error among 10 replicates. The black curve indicates a best-fit linear function calculated using Microsoft Excel. (**E**) Measured dependence of ring radius on the droplet size. Average mCherry ring radii obtained for microcolonies growing in 5-μl, 10-μl, and 15-μl droplets, respectively. These values were obtained from replicates at 36-h time points for 5-μl and 10-μl droplets and from replicates at 48-h time points for 15-μl droplets (only one mCherry ring emerged for this condition at the 36-h time point). All of the mCherry ring radii were calculated in the same manner as described in Figure 1. The error bars for 5-, 10-, and 15-μl droplets represent standard error among 3, 6, and 6 replicates, respectively. Cropped representative mCherry images of microcolonies to scale growing in 5-μl (top left panel) and 15-μl droplets (bottom right panel) at 36-h and 48-h time points, respectively, are shown. The color scheme is defined as in Figure 1.

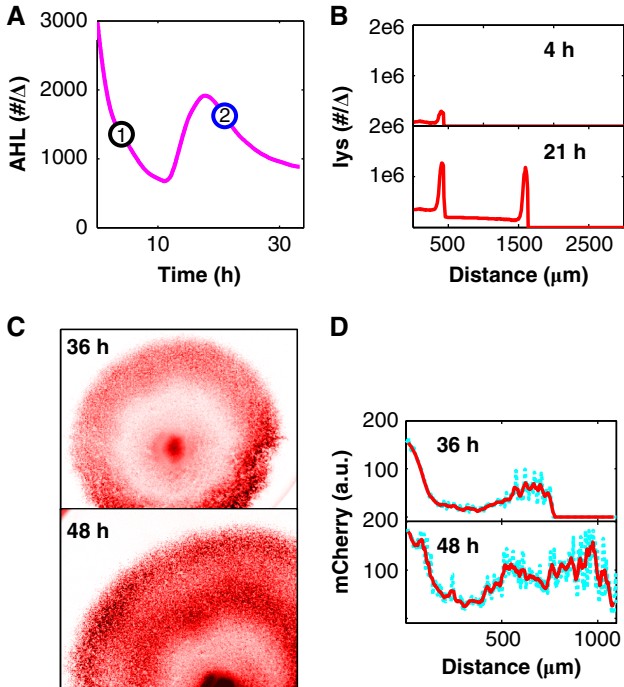

**Figure 4** Predicted and measured double-ring formation. (**A**) Simulated AHL dynamics for the double-ring case. AHL in molecules per $\Delta$ over time for the double-ring case simulation. Here, the simulation is implemented with the parameters listed in Supplementary Table S1. The initial AHL concentration in the simulation is 3000 molecules per $\Delta$. AHL crosses the threshold necessary for ring formation at two discrete time periods. (**B**) mCherry intensity (red line) at varying radii for time points 1 and 2 in **A**). The line plots indicate T7 lysozyme in molecules per $\Delta$ for the two time points indicated in (**A**) (4-h and 21-h time points, respectively). A single ring forms at ~50 $\Delta$ (~500 µm) for the 4-h time point. This ring is also maintained for the 21-h time point. However, after a second instance of crossing the AHL threshold necessary for ring formation, another ring at ~150 $\Delta$ (~1500 µm) emerges at the 21-h time point. (**C**) An experimentally obtained mCherry double-ring pattern. These images were obtained after culturing a single microcolony for 36 (top) and 48 (bottom) h at an initial AHL concentration of 100 nM. The color scheme is as described in Figure 1. (**D**) mCherry intensity (cyan dots) at varying radii for the images in (**C**). The solid red line is the running average at varying radii. mCherry was calculated as described in Figure 1 across angles spanning $5\pi/6$ to $3\pi/2$ for the 36-h time point and across angles spanning $\pi$ to $7\pi/6$ for the 48-h time point.

(Difco Laboratories), and 5 g NaCl were added to 1 l deionized $H_2O$. The 2xYT medium was then buffered to pH = 6.5 with KOH solution in 20.92 g/l MOPS (Omnipur, ⩾99%).

## Plasmids and cell strains

Our circuit consists of two plasmids: pET15bLCFPT7 and pTuLys2 CMR2. pET15bLCFPT7 was constructed as described in Tan *et al* (2009). This plasmid contains the activation module (Supplementary Figure S1, green dashed box). pTuLys2 CMR2 was constructed from the parent plasmids pLuxRI (Balagadde *et al*, 2005), pLuxmCherry (Collins *et al*, 2006), pET15bLCFPT7, and pLysS (Novagen, Madison, WI). pLuxmCherry and pET15bLCFPT7 were used as templates for PCRs, which produced two DNA segments containing the *pluxI* promoter upstream of mCherry and LacI, respectively. The two segments were joined using overlapping PCR. The 5′ primer used to amplify the *pluxI* promoter and mCherry contained an overhanging 16-base pair region constituting the *pT7* promoter in the reverse orientation relative to the *pluxI* promoter. The resulting PCR segment had EcoRI and AatII overhangs at the 5′ and 3′ ends, respectively. Both the PCR segment and *pluxI* were digested with EcoRI and AatII and

ligated to yield pTuLac2 with kanamycin resistance and a p15A origin of replication (ori). The Kan[r] gene was then removed and replaced with a Cm[r] gene by digesting pTuLac2 and pLuxGFPuv (Cm[r]) with SpeI and AatII and ligating the appropriate segments together. The resulting plasmid was named pTuLac2 CMR2. T7 lysozyme was then amplified from pLysS using PCR with a 5′ primer containing an AatII overhang and a 3′ primer containing an NheI overhang (all primers used in this study can be found in Supplementary Table S2). Upon digestion of pTuLac2 CMR2 and the PCR product with NheI and AatII, a ligation reaction was performed to yield the final plasmid, pTuLys2 CMR2. This plasmid contained the inhibition module (Supplementary Figure S1, red dashed box). Both plasmids were sequence verified.

The cell chassis used in this study is MC4100Z1 (gift of Michael Elowitz), which is MC4100 [*araD139 (argF-lac)205 flb-5301 pstF25 rpsL150 deoC1 relA1*] with a Z1 cassette for lacIq, tetR, and spect(R). For the activation module characterization, BL21 DE3 and BL21 DE3 pLysS cell strains (Novagen) were used.

## Activation module characterization

The activation module was tested for inhibition in response to T7 lysozyme. This test was conducted by comparing the CFP expression of cells in liquid culture containing the activation module plasmid (Supplementary Figure S1, green dashed box) in a BL21 DE3 cell chassis with or without the pLysS plasmid in response to a varying input of IPTG concentration. The BL21 DE3 cell chassis expresses T7 RNAP from a lacUV5 promoter, which is inducible by IPTG. The pLysS plasmid constitutively expresses T7 lysozyme. At each IPTG concentration, the mean CFP expression level of the cells containing pLysS is dramatically lower than that of the cells without the plasmid (Supplementary Figure S2). This result indicates that the T7 lysozyme expressed from the pLysS plasmid is indeed decreasing T7 RNAP expression as designed. Furthermore, this effect was observed across a large span of IPTG induction levels, indicating that inhibition of the activation module occurs over a wide range of gene expression levels. This result is again consistent with the circuit design. Additional characterization of the activation module (e.g., demonstration of activation by T7 RNAP and IPTG induction) was conducted previously (Tan *et al*, 2009).

## Experimental set-up

For Figures 1, 3C, and 4 and Supplementary Figures S4, S6, S14, and S15, overnight LB cultures of our engineered cells supplemented with 75 µg/ml carbenicillin and 50 µg/ml chloramphenicol were prepared and diluted to ~1–4 cells in 5 µl of 0.07% 2xYT (pH = 6.5) soft agar. The soft agar was supplemented with the same concentration of antibiotics and 1000 µM IPTG. Eight 5-µl aliquots of the soft agar mix were placed into 8 1-mm deep, 6-mm diameter wells of a CultureWell™ multiwell chambered coverslip (Grace Bio-Labs, Bend, OR, USA; item #103380), and a glass coverslip was applied to the top (see Supplementary Figure S3).

For Figure 3E, the same process was done for 5-µl, 10-µl, and 15-µl soft agar droplets with ~1–4 cells per droplet. However, four aliquots of soft agar mix were placed into 4 1-mm deep, 9-mm diameter wells of a CultureWell multiwell chambered coverslip (Grace Bio-Labs; item #103340).

## Mathematical modeling

We model the experimental pattern-formation system with an agent-based model in one spatial dimension. The main goal of this simplified model is to demonstrate the generic feasibility of pattern formation through a timing mechanism controlled by a uniformly distributed morphogen (i.e., AHL). The model captures the key aspects of the experimental system: interactions between major circuit components, as well as cell proliferation and movement.

In our model, cells are treated as individual agents within which intracellular reactions, namely the production and degradation of T7 RNAP and T7 lysozyme, as well as the production of AHL, take place. Whereas the T7 RNAP and T7 lysozyme proteins are confined to their

cell of origin, AHL freely diffuses across the cell membrane, and its degradation occurs both inside and outside the cell. We assume that gene expression capacity decreases as the distance between the host cell and the edge of the colony increases (Mcmichael, 1992), and we model this spatial dependence by means of a steep Hill function. More precisely, only a boundary layer of cells located close to the edge of the colony can efficiently synthesize proteins. T7 RNAP enhances its own production through a positive-feedback loop, and activates the production of AHL. AHL then induces lysozyme expression, which in turn inhibits T7 RNAP production. T7 RNAP and lysozyme are known to bind and form the *T–L* complex (denoted hereafter as P), and the corresponding reversible first order kinetics takes place on a very fast time scale (Kumar and Patel, 1997). Additionally, *T–L* complex can inhibit T7 synthesis on the transcriptional level as described in Jeruzalmi and Steitz (1998). In summary, the chemical reactions are captured by the following differential equations:

$$\text{T7 RNAP}\,(T): \frac{dT}{dt} = \left(\frac{K_P}{P + K_P}\right)\left(\frac{k_T T}{T + K_T}\right) - d_T T - k_{TL} TL + d_P P,$$

$$\text{Lysozyme}\,(L): \qquad \frac{dL}{dt} = \frac{k_L A^m}{A^m + K_A^m} - d_L L - k_{TL} TL + d_P P,$$

$$TL\,\text{complex}\,(P): \quad \frac{dP}{dt} = k_{TL} TL - d_P P,$$

$$\text{AHL}\,(A): \frac{\partial A}{\partial t} = D_A \Delta A + \left(\frac{K_P}{P + K_P}\right)\left(\frac{c k_A T}{T + K_T}\right) - d_A A,$$

where

$$k_T = \phi(r)k_{T_0},\; k_A = \phi(r)k_{A_0},\; k_L = \phi(r)k_{L_0},$$

$$\phi(r) = \frac{\left(\frac{1}{r}\right)^\gamma}{\left(\frac{1}{r}\right)^\gamma + \left(\frac{1}{K_\phi}\right)^\gamma} + \phi_0,$$

and *r* is the distance between the cell where the reactions take place and the edge of the 'cell colony' (the position of the cell furthest from the left end of the 1D grid, see below). On the basis of the fast diffusion of AHL, we assume instantaneous diffusion of AHL by setting $D_A = \infty$. Furthermore, the *T–L* complex is treated as a quasi-stationary field based on its fast reaction dynamics. The effect of changing $\phi$ by the multiplicative factor $\alpha$ is a corresponding decrease in the lysozyme (mCherry) ring radius (see Supplementary Figure S16). The multiplicative factor $\alpha$ is set to one for every simulation but those in Supplementary Figure S16.

The simulation is performed on a 1D grid of length *L*, with grid elements of length $\Delta = 10\,\mu m$. As explained above, the 1D framework is based on the radial symmetry of the experimental system, and the computational domain should be interpreted as a radial section of the expanding colony with the leftmost grid element corresponding to its center. Here, we impose the assumption that all cells within the same grid element have identical intracellular contents (i.e., they harbor identical *T* and *L* concentrations). Consequently, the equations for *T* and *L* are solved locally in each grid element, provided that at least one cell is present. On the other hand, the equation for A is solved over the entire spatial domain. *T* and *L* have units of #/cell. A has units of #/$\Delta$. As an initial condition, one hundred seeding cells are placed uniformly throughout the 10 leftmost grid elements at a concentration of 10 cells per element. Each seeding cell initially contains 100 T7 RNAP molecules and 1 lysozyme molecule.

The simulation of cell dynamics is built upon a previously established framework (Ferreira *et al*, 2002; Mallet and De Pillis, 2006), and is carried out based on a set of rules chosen empirically to capture the proliferation and jump dynamics of the cells. For each time step $\Delta t = 1$ min, one of two potential actions can be taken for each grid element that harbors at least one cell: division or movement, chosen randomly with equal probability. If division is the chosen event, then a new cell is generated with probability

$$1 - \exp\left[-\left(\frac{c}{\left(\frac{L}{L + K_{Lg}}\right)\left(\frac{T}{T + K_{Tg}}\right)\theta_g}\right)\right],$$

increasing the cell number from *c* to $c + 1$ in the respective grid element. Consequently, the intracellular contents summed up over *c* cells (before division) are distributed equally among $(c + 1)$ cells. If movement is the chosen event, then one cell inside the grid element moves with probability

$$1 - \exp\left[-\left(\frac{c}{\left(\frac{L^{h_L}}{L^{h_L} + K_{Lm}^{h_L}}\right)\left(\frac{T^{h_T}}{T^{h_T} + K_{Tm}^{h_T}}\right)\theta_m}\right)\right]$$

into one of the two adjacent grid elements. If both adjacent grid elements have fewer cells than the central element, then the moving cell chooses one of the two elements to move into with equal probability; if only one adjacent element has fewer cells, then the moving cell moves into that element; if neither element has fewer cells, then no jump takes place. The cells in the leftmost grid element can only jump to the right. As long as a jump occurs, the cell number in the original grid element is reduced by one. Consequently, the total number of each intracellular molecular species summed over the *c* cells (before movement) in the destination grid element and the moving cell is divided equally among the $(c + 1)$ cells (after movement) in the destination grid element; the intracellular contents (*T* and *L* per cell) of the remaining $(c − 1)$ cells in the original grid element are unchanged. Note that both the division and jump probabilities decrease with increasing intracellular lysozyme and T7 RNAP concentration. $\theta_m$ and $\theta_g$ are parameters controlling the baseline motility and growth potentials.

To constrain our model, we use literature values where available (see references in Supplementary Table S1), and choose the remaining parameters (within a reasonable range) in such a way that the simulated dynamics capture the salient features of the experimentally observed system. We expect that the critical role played by the uniformly distributed morphogen can be coupled to a wide array of intracellular and intercellular chemical and physical interactions to generate sophisticated yet controllable patterns. The exact pattern may depend on the specifics of the chemical and physical interactions, yet the controllability of the pattern actuated through the uniformly distributed morphogen is generally applicable and not restricted to our specific circuit.

## Data processing

For the experimental data presented in Figures 1, 3, and 4 and Supplementary Figures S4, S5, S6, S7, S14, and S15, the RFP imaging was done with the excitation filter set to 546/12 and the emission filter set to 605/75. The CFP imaging was done with the excitation filter set to 436/20 and the emission filter set to 480/40. The exposure levels were chosen independently for each image to avoid saturation, except in the case of Figure 1, Supplementary Movies 1 and 2, and Supplementary Figure S6, where the exposure settings were kept constant.

An edge detection algorithm was used to segment the core of the microcolony based on the phase image. The center of the microcolony was then calculated as the centroid of the microcolony core. For Figure 1, the intensity values were calculated by averaging across all angles spanning 0 to $2\pi$ after subtracting for background. For Figures 3 and 4 and Supplementary Figure S14, the intensity values were then calculated by averaging across all angles of a sector at fixed radii after subtracting for background. mCherry ring radius was defined as the radius corresponding to the maximum running average of mCherry. Sectors were chosen to avoid edges of the droplet and interference from other microcolonies growing in the same well. For Figure 3E, some microcolonies were so large such that a microcolony core did not appear within the frame of the image. In these cases, another algorithm was used to quantify mCherry ring radius. The algorithm calculated the running average of mCherry across the whole image, excluding

edges of the droplet. Then, the maximum running average of mCherry was indexed by location for each horizontal row in the image. A function relating this location to vertical distance was fitted to an equation for a perfect circle. On the basis of this best-fit function, the mCherry ring radius was extrapolated. All of these data analyses were conducted using custom code in MATLAB. In addition, single microcolonies per well were excluded from the data analysis process since their dynamics were significantly different in that the mCherry rings emerged later and were of a larger size than other samples under the same condition (see Supplementary Figure S15). Furthermore, microcolonies were initially excluded from being imaged if they rested too close to the edge of the droplet or were too close in proximity to another microcolony. In total, for Figure 3C, seven, ten, nine, and five samples were analyzed for the 0, 10, 30, and 100 nM AHL data points, respectively. For Figure 3E, three, six, and six samples were analyzed for the 5-μl, 10-μl, and 15-μl droplet data points, respectively.

All of the processing of the simulation results was done in the same manner described above for the experimental data. However, all molecule and cell numbers were taken directly from the 1D spatial domain as opposed to being averaged across angles.

## Supplementary information

## Acknowledgements

We thank J Wong, J Tabor, C Tan, and Y Tanouchi, and You laboratory members for discussions or comments on the manuscript; Duke IGSP's DSCR for assistance with high-performance computations; B Munsky, J Ozaki, H Song, and M Wall for preliminary analysis on earlier versions of the mathematical model; K Gonzales for discussions on the modeling framework; Y Gao and S Johnson for assistance with confocal microscopy; Duke Light Microscopy Core Facility (LMCF) for access to confocal microscopes and imaging software; M Elowitz for strain MC4100Z1; and C Collins for plasmid pLuxmCherry. This study was partially supported by the Office of Naval Research (N00014-12-1-0631), National Institutes of Health (LY: 1R01-GM098642; MDR: R01-GM096190-02), a DuPont Young Professorship (LY), a National Science Foundation CAREER award (LY), a David and Lucile Packard Fellowship (LY), an NIH/NIGMS Biotechnology Predoctoral CBTE Fellowship (to SP), and a DHS Graduate Fellowship (to SP). The research was performed under an appointment to the Department of Homeland Security (DHS) Scholarship and Fellowship Program, administered by the Oak Ridge Institute for Science and Education (ORISE) through an interagency agreement between the US Department of Energy (DOE) and DHS. ORISE is managed by Oak Ridge Associated Universities (ORAU) under DOE contract number DE-AC05-06OR23100. All opinions expressed in this paper are the author's and do not necessarily reflect the policies and views of DHS, DOE, or ORAU/ORISE.

*Author contributions:* SP and YC performed the experiments; SP performed the data analysis; BL and SP performed the mathematical modeling; SP, BL, and LY analyzed results and wrote the manuscript; DS and MDR provided critical inputs in model development and analysis.

## Conflict of interest

The authors declare that they have no conflict of interest.

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
