## [Review Process File · Molecular Systems Biology]

Temporal control of self-organized pattern formation without morphogen gradients in bacteria

Stephen Payne, Bochong Li, Yangxiaolu Cao, David Schaeffer, Marc D. Ryser, Lingchong You

Corresponding author: Lingchong You, Duke University

Review timeline:

Submission date:	03 April 2013
Editorial Decision:	03 May 2013
Revision received:	09 July 2013
Editorial Decision:	23 August 2013
Revision received:	02 September 2013
Accepted:	06 September 2013

Editor: Maria Polychronidou

Transaction Report:

1st Editorial Decision

03 May 2013

Thank you again for submitting your work to Molecular Systems Biology. We have now heard back from the four referees who agreed to evaluate your manuscript. As you will see from the reports below, the referees find the topic of your study potentially interesting. They raise, however, substantial concerns on your work, which should be convincingly addressed in a major revision of the manuscript.

In particular, the reviewers point out that additional experimentation is required in order to convincingly support the proposed mechanism of pattern formation. In addition to several important control experiments that should be included as suggested by the reviewers, two major points that need to be addressed are the following:

- Direct evidence needs to be provided in order to demonstrate the existence and the role of metabolic heterogeneity in pattern formation.
- The robustness of the pattern formation and its dependence on the synthetic circuit should be demonstrated more convincingly.

On a more editorial level, we would like to encourage you to include the source data for the figures that show essential quantitative data.

If you feel you can satisfactorily deal with these points and those listed by the referees, you may wish to submit a revised version of your manuscript. Please attach a covering letter giving details of the way in which you have handled each of the points raised by the referees. A revised manuscript will be once again subject to review and you probably understand that we can give you no guarantee at this stage that the eventual outcome will be favorable.

 Referee reports:

Reviewer #1:

'Temporal control of self-organized pattern formation without morphogen gradients in bacteria' is an important contribution with potential wide-ranging implications well beyond synthetic biology. The coupling of metabolism with pattern formation has a long history, beginning with the pioneering work of Charles Manning Child at the turn of the 20th century (eg. C. M. Child 'Patterns and problems of development' University of Chicago Press 1941; G. H. Parker (1929) 'Metabolic gradient and its applications' Journal of Experimental Biology 6: 412-426). Child's work was largely cast aside after his death, coincidental with the publication of Turing's famous work on pattern-forming instabilities. Payne et al. show that this long-neglected coupling between physiological state and patterning has a place in modern biology.

The authors validate the role of AHL convincingly, but the role of global gene expression coupled to metabolism (which is central to their interpretation of the results) is left implicit. There is an established quantitative relationship between nominal nutrient quality and gene expression levels (Klumpp et al., 2009) - to substantiate the proposed mechanism (and the model parameterization), the experiments in Fig 1B should be repeated with a variety of nutrient agar (in addition to the LB supporting about 2.5 dbl/h). A good range would be 0.4% glucose minimal MOPS media (supporting about 1 dbl/h) and 60 mM acetate minimal MOPS media (supporting about 0.3 dbl/h). That should change the basal k_T , k_L and k_A synthesis rates according to Klumpp. Without this additional validation, the proposed mechanism is unsubstantiated. Although the results as they stand are interesting to the synthetic biology community, the work will be of interest to a much wider audience if the coupling between metabolism, gene expression and patterning can be established convincingly.

Other comments:

1. The term 'morphogen' should be defined. The authors have in mind a broader definition than is common and that ought to be made clear.

2. The simulation of cell dynamics is based on tumor cell modeling. It is not obvious that motility should play a role in the dynamics of E coli. Do the authors envision chemotaxis driving that movement? My understanding is that the host strain MC4100Z1 carries a *flb-5301* mutation that abrogates flagellar synthesis.

Is it a typographical error that the motility term (bottom p. 16) exhibits sharp Hill-type kinetics (through h_L and h_T) whereas the division term exhibits a weaker hyperbolic dependence? If that is not a typographical error, it should be explained in some detail. The impression that is left is that spatial-dependent motility - and not spatial-dependent gene expression - is what underlies the patterning.

It would be helpful to the reader if the simulation code was made available through the supplementary materials.

Reviewer #2:

In Payne et al., the authors develop an activator-repressor circuit. The activator consists of an autocatalytic T7 RNA polymerase. T7 RNAP also activates transcription of an inhibitor module. The inhibitor module consists of *luxR*, *luxR*, and T7 Lysozyme. *luxR* and *luxI*. *luxI* produces a diffusive protein AHL (acyl-homoserine lactone). AHL activates the *LuxR* protein to enable transcription of T7 Lysozyme from a *luxI* promoter. T7 Lysozyme then inhibits the production of T7 RNAP.

The naive expectation for this circuit would be the enhancement of any initial T7 production foci through positive feedback with inhibition of T7 production outside of these foci depending on the parameters for kinetics and diffusion. The authors, however, discover an annular pattern outside of

an initial focus of T7 production. They attribute the spatial pattern to a metabolic state and explore the pattern through fluorescence microscopy experiments and agent based simulation. The concept described in this work is interesting, but as is, the paper lacks numerous critical controls and too many assumptions are made without any direct experimental support.

Below are specific questions and comments:

1) Is the the spatial distribution of the metabolic pattern intrinsic to MC4100Z1 or is it an unintended effect of the synthetic genetic circuit? The authors suggest contact inhibition, mechanical stress, and local nutrient concentration as possible reasons for the apparent spatial metabolic pattern. They, however, also discuss the metabolic burden induced by production of T7 Lysozyme and T7 RNAP. The pause in colony growth for example is attributed to the metabolic burden of their genetic circuit. Since the metabolic pattern is central to the problem being explored, this is critical to clarify. The authors must present some direct evidence for the claims regarding heterogeneity in metabolic activity.

a. Is there a perturbation that can eliminate the annular metabolic pattern? The authors have developed computational and experimental tools to evaluate the observed pattern. However, they do not show any condition where the pattern can be completely removed or otherwise state that this is not possible. This would help show they understand a causative relationship for the pattern.

b. Can the authors demonstrate the annular metabolic pattern more directly in the absence of the activator-repressor synthetic genetic circuit? If the metabolic pattern is not due to the circuit, then the authors should be able to demonstrate the spatial metabolic pattern in the absence of the synthetic circuit. A minimal reporter system for the observed metabolic pattern would be helpful to understand whether the pattern is intrinsic to the strain or due to the circuit.

c. Is the annular ring due to cell density or the 2D projection?

d. How does the expression of T7 RNAP or lysozyme change the growth rate?

e. Are there any other factors (such as nutrients), which could generate metabolic activity patterns?

2) What is the role of LuxR in the annular pattern? AHL activates LuxR in order to induce production of mCherry and T7 Lysozyme. This creates an AND gate like pattern between LuxR and AHL. If LuxR is not present in the cell, would there be an mCherry production? Is LuxR production linked to the metabolic state?

3) The the 1D concentrations be related to physical units of molarity for comparison between experimental and computational results? The use of μm ($\approx 10 \mu\text{m}$) for the 1D spatial unit makes the paper more difficult to relate to physical parameters. Using μm in the figures and their legends would be make it more clear. Also could the 1D simulation be interpreted to be analogous to an average over all angles radiating from the center of the colony. Assigning a physical interpretation

4) A major discrepancy between the model and experiments is the mCherry signal at the center of colonies. While strong mCherry signals are observed at the center and the ring in the experiments, the model only accounts for the ring patterns. What causes this discrepancy?

5) The authors excluded single microcolonies per well for their analysis, because their dynamics were significantly different (page 18). However, the difference explained by the authors seems to support their theory. The model would predict larger rings that appear later would be caused by lower initial cell numbers (since lower initial cell density would take longer time to accumulate AHL). How would the number of initial cells affect the dynamics? The size of the ring should correlate with initial cell number. Is this relationship something the model would predict? If not, why?

6) Another critical assumptions of the model is stated by the authors on page 6: "the cell's global gene expression decreases with increasing distance from the microcolony edge". Because this is the key for the pattern generation, strong experimental support should be present.

7) How stable is AHL in under the experimental condition used in this study? Is there any way to measure (or estimate) the actual concentration of AHL? The authors claimed "AHL concentration gradually decreases due to reduced T7 RNAP strength (Figure 2A) (page8)", but this is only true

when AHL degrades. If AHL is very stable, termination of production does not decrease the concentration. Experimental support for the degradation of AHL should be present.

8) The brightness of mCherry signal on the ring does not look homogeneous. Some part of the ring looks brighter than the other part of the ring. How do the authors account for this observation?

9) What is the "dip" in the colony radius plot in fig1D? There seems to be a decrease in colony radius around 30hr. Is this "dip" real or some kind of artifact?

10) An interesting prediction of the model is the two phase growth behavior during colony expansion. The colony radius initially increases rapidly, and increases again after slow expansion period (Figure 1F). Does this agree with the experimental results? If so, how does this fit in the size control mechanism the authors discussed in page 11?

11) The two rings in Fig4c are not very clear to me. Better representative images should be shown. Also, why does the 1st ring seem to be shifted to the inside from 36hr (around 700um) to 48hr (500um)?

12) Adding control experiments with the strain which only carries an activation or inhibition module would make enhance the strength of the story.

Major Presentation Point

The x-axis representing time for Figure 1f should be extended to match Figure 1d for comparison between computational and experimental results.

Minor Presentation Points

The figure quality needs to be improved for publication. Specifically:

- 1) The solid lines red and blue lines are too thick causing the following issues:
 - i. In Figure 1c obscure the primary data indicated by the blue and green dots that appear to be actually dashed lines. If the dots are actually dashed lines, the figure legend should be corrected.
 - ii. It is difficult to see variation of the mCherry ring location in 1d
 - iii. It is difficult to see what appears to be noise of the red and green lines in 1b
- 2) The location of the fluorescent reporters in the circuit not very clear from Figure 1a
- 3) Figure 1b should have a color bar for better quantitative representation and indication of the saturation and would be clearer if an additional visual aide were given for the time progression from left to right. The relationship of the coloring to Figure 1 could be made more clear.
- 4) Inclusion of the y-axis label in Figures 2a-d would be helpful. The split y-axis between the three time points in panels b and c make quantitative comparison difficult. Figure 2d is particularly not clear without axis labels. For the AND gate, more explicit labeling of the gate would be helpful.
- 5) Comparison of the three magenta traces in Figure 3a is difficult.
- 6) Two time points are indicated in 3B which seems to correspond to different AHL concentrations

Minor Points

- 1) The strain MC4100Z1 should be described in more detail or a reference should be given with such a description.

Reviewer #3:

In their manuscript "Temporal control of self-organized pattern formation without morphogen gradients in bacteria," Payne et al. report a synthetic gene circuit that generates a ring pattern in the apparent absence of a morphogen gradient. Synthetic gene circuits that give rise to similar patterns have been demonstrated previously, but they have relied on spatial inhomogeneity of intercellular signaling molecules to drive pattern formation. In contrast, the authors argue that their circuit generates a stable pattern based on the temporal dynamics of a spatially homogeneous intercellular

signal (AHL). Combined with a spatial inhomogeneity in overall gene expression and cell growth, a global increase in AHL initiates a positive-feedback loop that maintains the pattern even as the microcolony continues to grow.

The authors argue convincingly that their system's patterns form independently of a morphogen gradient. Their agent-based model successfully recapitulates the major features of the system, predicting several behaviors that are experimentally verifiable. The work is exciting and has significance both for efforts to understand natural pattern formation and for engineering them in synthetic systems, and I think it deserves publication. However, there are several points that could make their argument stronger and easier to follow.

Major comments:

1. The initial experimental evidence that pattern formation proceeds in the absence of an AHL gradient rests on the exogenous application of 100 nM AHL; however, not enough context is given to evaluate the claim that this effectively abolishes any extant AHL gradient. While adding AHL obviously changes the relative strength of a hypothetical spatial gradient, the authors in fact still saw an effect: a smaller pattern. I don't think that additional experiments are called for, but putting this experiment in better context would make it more convincing. For example, is there a reference that could be cited where an AHL gradient does modulate a spatial effect, and where an additional 100 nM abolishes that effect?
2. One of the strengths of model-based circuit characterization is in identifying non-obvious properties of the system that are crucial for producing the observed (or desired) behavior. This is exactly what happened when the authors observed that spatially dependent variation in global gene expression was essential for their model to recapitulate the observed phenomena. It's not clear, however, whether "increased global gene expression", "faster cell growth" and "high metabolic capacity" at the edge of the growing microcolony refer to the same phenomenon, or different but coupled phenomena (cf. Scott et al, Science 330:1099 (2010)). A better explanation of the relationship between these terms (concepts? phenomena?) would strengthen the argument for including a seemingly arbitrary model component.
3. Finally, the authors propose that the location of lysozyme (and RFP) expression is determined by what amounts to a logical AND gate between AHL level and the cells' metabolic capacity (Figure 2D). This conceptual model is neat, logical and accounts for the behavior they observe; but it's not clear how well their results support it. Does varying the AND gate's other input (metabolic capacity), either in simulation and/or experiment, also modulate the pattern size? Alternately, is there a direct measure of metabolic capacity (perhaps an integrated, constitutively expressed fluorescent marker) that could be used to bolster the assertion that metabolic capacity is higher on the outer edge of the growing microcolony?

Minor corrections:

1. Is a multi-well coverslip really a "microchannel device"? To my mind, "microchannel" implies that something is flowing through it.
2. I found the Y-axis labels in Figure 2 to be terribly confusing: the measurement being indicated appears to be at the top of the plots (or nowhere at all, as in D), and there are no units (except in C). Also, in the stacked plots (B and C), only a single tic label is given for each plot, so it's not immediately obvious why there's a "70" or a "3" floating next to the plot. Please either label the Y axes more completely; or, if the units are unimportant, indicate that the scale is arbitrary and omit them entirely.
3. I found your timelapse movies quite striking. If you have one for the double-ring experiment (and there's room left in your SI allocation), you might consider including it.
4. For Figures S2 and S3, you might consider moving the experimental protocols from the figure legends into a separate block of SI text.
5. For the experiment described in Figure S2 and the accompanying legend, please describe pLysS.

It was not clear until I looked it up that it expresses lysozyme under control of a lac operator.

Reviewer #4:

In this manuscript, the authors tried to demonstrate that the self-organized pattern formation could be achieved independent of the morphogen gradient. They employed an engineered *E. coli* strain that carrying two plasmids (circuits), an activator and an inhibitor, as an experimental model for pattern formation analysis. They presented that the ring patterns generated by the *E. coli* colonies (reported by mCherry) formed in soft agar were not determined by a spatial cue due to the morphogen gradient, but were self-regulated in a timing cue. The authors concluded that a novel mechanism on pattern formation was demonstrated, according to their experimental evidence along with the theoretical simulation. The idea/model on pattern formation mechanism is nice and highly appreciated, but I do feel regret that the results presented in this study could not persuade me to fully accept their conclusion. Briefly, two major points seriously prevented me to support the authors' statement.

1. It is doubted whether the results fully supported the conclusion. It is true that AHL was the only controlled morphogen in the study, but it didn't mean only a single signal/morphogen determine the pattern formation. Other elements, which were not detected or evaluated in this study, for instance, those involved in the so-called metabolic burden, possibly contributed to the ring pattern.
2. It seems that only several colonies were selected for data processing. I'm wondering if such a few data sets are sufficient for estimation. It's hard to say whether the observation on ring formation was statistically reliable. Also, no negative controls were provided, for instance, the controls of the cells carrying a single circuit, and/or the cells carrying the reversed genetic constructions, etc. I'm afraid the so-called "robust" (repeated in the manuscript) pattern formation does not sound so much "robust".

Minor comments:

1. The authors used the phrase "robust, self-organized pattern" repeatedly in the manuscript. What does the terms "robust" and "self-organize" precisely refer to?
2. p3, 1st paragraph, the last sentence: "...the generation of robust, self-organized pattern of gene expression in the absence of an apparent morphogen gradient" was confusing. It needs to be fixed.
3. Why a special device of microchips was used for the colony formation? Is it advantageous in this study? Is it possible that the ring pattern of the colonies was somehow device-dependent?
4. Why 1-4 cells were inoculated in each well/chamber? Are there any reasons in particular? Would the initial numbers of cells play a role in the ring formation?
5. The figure legends contained insufficient information. For example, what the "#" in Figs. 2 and 4 stands for? I'm not sure if I caught the proper meaning of the figures.
6. The authors are recommended to pay attention to the formats in the manuscript, such as, the inconsistent formats of references, the differences in the calligraphy between the figures (small letters) and the figure legends (capital letters).

1st Revision - authors' response

09 July 2013

We thank the reviewers for evaluating our work. We are glad that all reviewers have found our central conclusions novel and potentially significant. That is, by using a synthetic system, we have demonstrated that a morphogen can serve as a timing cue, instead of a spatial cue, to trigger the formation and maintenance of self-organized patterns. We note that this point has implications

beyond the molecular mechanisms underlying the ring pattern formation in our system. The reservations by the reviewers primarily focus on the potential mechanisms underlying the generation of the ring patterns in our system. In response to their suggestions and comments, we have thoroughly revised the manuscript text (shown in red) and included additional or updated data:

- 1) Direct experimental demonstration of ring formation in a different strain (MG1655) in a different experimental platform to further illustrate the robustness of the pattern formation process (Fig. S5).
- 2) Experimental data demonstrating the lack of an mCherry ring pattern for MC4100Z1 *E. coli* strains containing the full circuit with LuxR or LuxI knocked out (Fig. S6).
- 3) Direct experimental evidence to demonstrate the existence of metabolic heterogeneity in the form of gene expression capacity via an MC4100Z1 *E. coli* strain with an inducible fluorescent reporter (Fig. S7).
- 4) Fluorescent image of an additional double-ring pattern (Fig. S14).
- 5) Fluorescent images of agar droplets containing a single microcolony (Fig. S15).
- 6) Mathematical modeling of perturbations to the gene expression capacity term (Fig. S16).
- 7) The source data for the figures that show essential quantitative relations.
- 8) The source code for modeling of the system and data analysis.

With these new data, clarifications, and corresponding revisions, we hope you and the reviewers will find the manuscript sufficiently improved and acceptable for publication.

Point-by-point responses to editorial and reviewer comments

Editorial comments

Thank you again for submitting your work to Molecular Systems Biology. We have now heard back from the four referees who agreed to evaluate your manuscript. As you will see from the reports below, the referees find the topic of your study potentially interesting. They raise, however, substantial concerns on your work, which should be convincingly addressed in a major revision of the manuscript.

In particular, the reviewers point out that additional experimentation is required in order to convincingly support the proposed mechanism of pattern formation. In addition to several important control experiments that should be included as suggested by the reviewers, two major points that need to be addressed are the following:

- Direct evidence needs to be provided in order to demonstrate the existence and the role of metabolic heterogeneity in pattern formation.

Indeed, a critical requirement for the initiation of ring patterns is the spatial dependence of gene expression in a microcolony: cells near the edge have higher gene expression capacity than those in the interior of the microcolony. As noted in our manuscript (page 7), this assumption is fully consistent with our current understanding of bacterial physiology. The reduced metabolic capacity in the interior could be due to a number of factors, including mechanical stress, accumulation of potentially toxic metabolites, cell-cell contact inhibition, or depletion of nutrients. Regardless of the underlying molecular mechanism, the critical issue is whether cells at the edge are able to express genes at a faster rate.

To directly test this notion (as also suggested by Reviewers #2 and 3), we used MC4100Z1 cells (the same *E. coli* strain as used throughout the study) expressing a fluorescent protein (mCherry) under the control of the inducible *ptet* promoter. This strain was placed in a soft agar droplet with the same experimental protocol used previously to obtain pattern formation with the synthetic gene circuit. The *ptet* promoter was induced with aTc (anhydrotetracycline), which could sequester the constitutively expressed inhibitor TetR. Examination of the spatiotemporal dynamics of mCherry expression in the microcolony indicates that at several different time points during the expansion of the microcolony, mCherry expression is higher at the edge of the microcolony (Fig. S7). Since mCherry is driven by a constitutively active *ptet* promoter in this system, the most plausible explanation for these results is that higher gene expression occurs on the edge of the microcolony.

We note that, while the ring can form from mCherry by a single inducible promoter, this ring is fundamentally different from that by our full circuit. The ring in this experiment expands along with the microcolony as it reflects the spatial dependence of the gene expression capacity of the cells. In contrast, the ring by the full circuit is triggered by a sufficiently high AHL (the timing cue) and a sufficiently high gene expression capacity. The ring then stays at a constant size despite subsequent colony expansion, and in this manner, ring size is decoupled from later microcolony expansion (Fig. 2).

- The robustness of the pattern formation and its dependence on the synthetic circuit should be demonstrated more convincingly.

Robustness is defined as the persistence of a system's characteristic behavior. We believe that the robustness of the pattern formation is aptly demonstrated in Fig. S4, where we show similar patterns arising for six independent clones, captured during 5 different, independent experiments. These patterns arise despite variations in experimental conditions, including droplet composition, agar density, and number of cells per droplet. Additional experiments with the same gene circuit in a different cell strain using a different experimental setup revealed a similar mCherry ring pattern (Fig. S5). Therefore, we consider the pattern formation arising from the gene circuit's dynamics to be robust.

In addition, in regard to the point questioning whether the pattern is indeed the result of the synthetic gene circuit, our new experiment described above (Fig. S7) demonstrates an mCherry ring coinciding with the continuously moving edge of the microcolony for MC4100Z1 transformed with a plasmid encoding aTc-inducible mCherry. These data indicate that the robust ring pattern of a constant radius achieved with MC4100Z1 containing the full circuit was due to the synthetic gene circuit's dynamics, which could be effectively altered by modulating the domain size and amount of initial, exogenously added AHL.

Furthermore, we have included new experimental data elucidating the spatiotemporal dynamics of two MC4100Z1 strains containing the full circuit with LuxR and LuxI knocked out, respectively (Fig. S6). For these two strains, a lack of a detectable mCherry ring is apparent. These experimental results directly demonstrate the essentiality of AHL signaling for the obtainment of the mCherry ring pattern. They also highlight that the pattern obtained in our study is critically dependent on the synthetic gene circuit.

Equally important, our model of cells containing the full circuit correctly predicts how the quantitative aspects of the ring patterns can be modulated experimentally (including the dependence of ring size on the domain size and the initial AHL concentration, as well as the generation of double rings). This provides another piece of critical evidence for the notion that the observed patterns are indeed due to our synthetic gene circuit.

On a more editorial level, we would like to encourage you to include the source data for the figures that show essential quantitative data. (Additional information is available in the "Guide for Authors" section in our website at <http://www.nature.com/msb/authors/index.html#a3.5.2>>)

We have added the source data for the figures that show essential quantitative relations. Specifically, we have included all of the raw fluorescent images constituting all of the data points in Figures 3C and 3E as well as all other raw fluorescent images used as data in the Main Figures. In addition, we have included the source MATLAB code used to run simulations and quantitatively characterize simulations and experimental data.

Reviewer #1:

'Temporal control of self-organized pattern formation without morphogen gradients in bacteria' is an important contribution with potential wide-ranging implications well beyond synthetic biology. The coupling of metabolism with pattern formation has a long history, beginning with the pioneering work of Charles Manning Child at the turn of the 20th century (eg. C. M. Child 'Patterns and problems of development' University of Chicago Press 1941; G. H. Parker (1929) 'Metabolic gradient and its applications' Journal of Experimental Biology 6: 412-426). Child's work was largely cast aside after his death, coincidental with the publication of Turing's famous work on

pattern-forming instabilities. Payne et al. show that this long-neglected coupling between physiological state and patterning has a place in modern biology.

We thank the reviewer for acknowledging the importance of the study as well as its implications in a broad area of fields outside synthetic biology. In particular, we thank the reviewer for noting the importance of the often-neglected coupling between a cell's physiological state and patterning. We have also cited the references suggested by the reviewer.

The authors validate the role of AHL convincingly, but the role of global gene expression coupled to metabolism (which is central to their interpretation of the results) is left implicit. There is an established quantitative relationship between nominal nutrient quality and gene expression levels (Klumpp et al., 2009) - to substantiate the proposed mechanism (and the model parameterization), the experiments in Fig 1B should be repeated with a variety of nutrient agar (in addition to the LB supporting about 2.5 dbl/h). A good range would be 0.4% glucose minimal MOPS media (supporting about 1 dbl/h) and 60 mM acetate minimal MOPS media (supporting about 0.3 dbl/h). That should change the basal k_T , k_L and k_A synthesis rates according to Klumpp. Without this additional validation, the proposed mechanism is unsubstantiated. Although the results as they stand are interesting to the synthetic biology community, the work will be of interest to a much wider audience if the coupling between metabolism, gene expression and patterning can be established convincingly.

We are glad that the reviewer is convinced of the proposed role of AHL, which is indeed the central conclusion of our study.

Regarding the role of gene expression, the critical assumption is that cells near the edge of a microcolony can express genes at a faster rate than those in the interior of a microcolony (everything else being equal). This notion is indeed consistent with our new data on the gene expression pattern from MC4100Z1 cells constitutively expressing mCherry: higher mCherry expression always occurs at the edge of the expanding microcolony (Fig. S7).

This spatial-dependent gene expression capacity can be due to several factors. For example, cells in the interior have lower gene expression capacity likely due to contact inhibition [1] and mechanical stress [2]. Alternatively, interior cells may suffer from lower nutrient concentration (as also noted by the reviewer) or higher concentrations of waste products that limit growth. It is unclear precisely which different factors contribute significantly to the overall gene expression capacity. In our particular system, however, we suspect that nutrient gradient had only minor (if any) contributions to the spatial-dependence of the gene expression capacity. In particular, the medium (2xYT) we used for the study is very rich. Within the time scale of the experiment, the nutrient is likely still at excess. Also, at the length scale of the microcolonies (~100-1000 μm) and the time scale of patterning (~30-70 hrs), the spatial gradients in nutrient concentrations (assuming the molecules are of similar size as AHL) are likely negligible, though we cannot completely eliminate the possibility of small gradients. Furthermore, the interplay between growth and circuit dynamics is absolutely critical in the pattern formation process, and with unknown perturbations to growth (by changing nutrient concentration), in addition to unknown perturbations to T7 lysozyme and T7 RNAP levels which influence growth, it would be very difficult to draw conclusions from an experiment perturbing nutrient level.

Most critically, we would like to emphasize that, as long as the spatial-dependence gene expression exists (it does), how it comes about is not critical for the interpretation of the ring pattern formation. Elucidating the molecular mechanisms is indeed an interesting problem for future studies.

Other comments:

1. *The term 'morphogen' should be defined. The authors have in mind a broader definition than is common and that ought to be made clear.*

We have added a definition to the term, "morphogen," on page 3. We use the following definition from Borello and Pierani: a signaling molecule produced by a localized source which activates a specific and distinct cellular response in a concentration-dependent manner [3].

2. *The simulation of cell dynamics is based on tumor cell modeling. It is not obvious that motility should play a role in the dynamics of E coli. Do the authors envision chemotaxis driving that*

movement? My understanding is that the host strain MC4100Z1 carries a *flb-5301* mutation that abrogates flagellar synthesis.

The reviewer is correct that MC4100Z1's flagellar pathway is inactive. For that reason, we do not envision chemotaxis driving cell movement. Instead, we believe that cell movement is driven primarily by random walk and by cell division. However, we note that the same circuit in a motile strain (MG1655), where chemotaxis could play a role in cell movement, also led to the formation of ring patterns (Fig. S5).

Is it a typographical error that the motility term (bottom p. 16) exhibits sharp Hill-type kinetics (through h_L and h_T) whereas the division term exhibits a weaker hyperbolic dependence? If that is not a typographical error, it should be explained in some detail. The impression that is left is that spatial-dependent motility - and not spatial-dependent gene expression - is what underlies the patterning.

It is not a typographical error. We should clarify that the goal of our model was to capture the essential, qualitative features of the system dynamics. These include the generation of a ring with constant size over time, the dependence of the ring size on initial AHL concentration and domain size, and the ability to generate double rings. The model was indeed able to capture all of these features.

Indeed the model does account for spatial-dependent motility, but it accounts for spatial-dependent gene expression as well. We have provided additional simulations perturbing the metabolic capacity ϕ with a multiplicative factor (which we called α) in Fig. S16. Here, ϕ represents spatial-dependent gene expression, and it is evident that by increasing ϕ with multiplicative factor α , the ring size decreases accordingly. Thus, spatial-dependent gene expression is indeed a critical factor underlying the patterning process.

It would be helpful to the reader if the simulation code was made available through the supplementary materials.

We have added simulation code in the revised submission.

Reviewer #2 :

*In Payne et al., the authors develop an activator-repressor circuit. The activator consists of an autocatalytic T7 RNA polymerase. T7 RNAP also activates transcription of an inhibitor module. The inhibitor module consists of *luxR*, *luxR*, and T7 Lysozyme. *luxR* and *luxI*. *LuxI* produces a diffusible protein AHL (acyl-homoserine lactone). AHL activates the *LuxR* protein to enable transcription of T7 Lysozyme from a *luxI* promoter. T7 Lysozyme then inhibits the production of T7 RNAP.*

The naïve expectation for this circuit would be the enhancement of any initial T7 production foci through positive feedback with inhibition of T7 production outside of these foci depending on the parameters for kinetics and diffusion. The authors, however, discover an annular pattern outside of an initial focus of T7 production. They attribute the spatial pattern to a metabolic state and explore the pattern through fluorescence microscopy experiments and agent based simulation.

The concept described in this work is interesting, but as is, the paper lacks numerous critical controls and too many assumptions are made without any direct experimental support.

We thank the reviewer for his or her interest in our work. We hope that our clarifications and additional experimental data will remove the concerns raised by the reviewer.

Below are specific questions and comments:

1) Is the the spatial distribution of the metabolic pattern intrinsic to MC4100Z1 or is it an unintended effect of the synthetic genetic circuit?

As is pointed out in response to the Editorial comments, the spatial distribution of the metabolic pattern (or spatial-dependent gene expression) is likely the intrinsic property of a growing MC4100Z1 colony. This is strongly suggested by the experiment described in Fig. S7. This same spatial-dependent gene expression is also likely at play in the *E. coli* strain MG1655, as we saw a similar pattern form when this strain was transformed with our full gene circuit in a different experimental setup (Fig. S5). We suspect that this differential gene expression is present in other *E. coli* strains as well, but future work is necessary to confirm this hypothesis.

The authors suggest contact inhibition, mechanical stress, and local nutrient concentration as possible reasons for the apparent spatial metabolic pattern. They, however, also discuss the metabolic burden induced by production of T7 Lysozyme and T7 RNAP. The pause in colony growth for example is attributed to the metabolic burden of their genetic circuit. Since the metabolic pattern is central to the problem being explored, this is critical to clarify. The authors must present some direct evidence for the claims regarding heterogeneity in metabolic activity.

We apologize for the confusion. Indeed, there are two aspects for the coupling between circuit components and cell physiology. In the microcolony, there is a spatial dependence in the gene expression capacity of cells: cells at the edge are able to express genes at a faster rate than those in the interior. This differential gene expression capacity could be due to differences in contact inhibition, mechanical stress, and local nutrient concentrations.

On the other hand, expression of T7 RNAP and T7 lysozyme indeed causes growth inhibition, which is distinct from the coupling described above. T7 RNAP's inhibitory effect on cell growth in the same cell strain (MC4100Z1) was documented previously [4]. T7 lysozyme's inhibitory effect on cell growth was documented in several previous studies [5-7]. We have added [5] as a reference in the revised manuscript.

The spatial dependence of gene expression capacity is consistent with our current understanding of bacterial physiology. Contact inhibition [1] and mechanical stress [2], which are likely to be more significant at the interior of microcolony, provide probable explanations for lower gene expression in those regions. Another possible contributing factor (but likely to a lesser degree) is that a cell's gene expression capacity is higher at the edge of the microcolony due to greater access to nutrients. This hypothesis is consistent with experimental evidence and the literature [8, 9].

As suggested by the reviewer, we have also carried out additional experiment to directly demonstrate this point. As shown in Fig. S7, a microcolony of MC4100Z1 cells expressing mCherry under an aTc-inducible promoter indeed exhibits higher mCherry expression at the edge of the growing microcolony for a number of different time points.

We note that, while the ring can form from mCherry by a single inducible promoter, this ring is fundamentally different from that formed by our full circuit. The ring in this experiment expands along with the microcolony as it reflects the spatial dependence of the gene expression capacity of the cells. In contrast, the ring by the full circuit is triggered by a sufficiently high AHL concentration (the timing cue) and a sufficiently high gene expression capacity. The ring then stays at a constant size despite subsequent colony expansion, and in this manner, ring size is decoupled from later microcolony expansion (Fig. 2).

a. Is there a perturbation that can eliminate the annular metabolic pattern? The authors have developed computational and experimental tools to evaluate the observed pattern. However, they do not show any condition where the pattern can be completely removed or otherwise state that this is not possible. This would help show they understand a causative relationship for the pattern.

We assume that the reviewer is referring to the ring pattern formed by the circuit. We note that the spatial dependence of the gene expression capacity of cells in a microcolony is one of the critical factors triggering the ring pattern. The other is the sufficient accumulation of AHL (Fig. 2).

The spatial dependence of gene expression capacity is likely an intrinsic property of a growing microcolony. For the cells carrying our full circuit, the formation of a ring with constant size (over time) is also a robust property. This property can indeed be perturbed in several ways. For example, the ring size can be predictably modulated by controlling initial AHL concentration (Figs. 3A-C) or domain size (Figs. 3A, D-E). Here, each perturbation affects the timing of AHL accumulation in an isolated and controllable manner *without affecting other variables*. We believe that these perturbation experiments have indeed demonstrated the causal relationship to which the reviewer pointed.

In light of reviewer's comments (including point 2 below), we have also carried out experiments with MC4100Z1 cells carrying two "mutant" circuits. Both are the same as the full circuit except that one does not contain LuxR and the other does not contain LuxI. These experiments were done using the same condition we used to observe the ring patterns shown in Figs. 1 and S4. Neither mutant circuits generated detectable mCherry rings (Fig. S6). These experimental results directly demonstrate the essentiality of AHL signaling for the obtainment

of the mCherry ring pattern. They also highlight that the patterns obtained in our study are indeed critically dependent on the synthetic gene circuit.

b. Can the authors demonstrate the annular metabolic pattern more directly in the absence of the activator-repressor synthetic genetic circuit? If the metabolic pattern is not due to the circuit, then the authors should be able to demonstrate the spatial metabolic pattern in the absence of the synthetic circuit. A minimal reporter system for the observed metabolic pattern would be helpful to understand whether the pattern is intrinsic to the strain or due to the circuit.

We thank the reviewer for the suggestion. We have carried out an additional experiment to demonstrate this point. As shown in Fig. S7, a microcolony of MC4100Z1 cells constitutively expressing mCherry indeed exhibits higher mCherry expression at the edge of the microcolony for multiple time points.

c. Is the annular ring due to cell density or the 2D projection?

If the reviewer is referring to the apparent mCherry ring in the fluorescent microscope images (Fig. 1b), the ring is due to higher local expression of mCherry.

d. How does the expression of T7 RNAP or lysozyme change the growth rate?

T7 RNAP's inhibitory effect on cell growth in the same cell strain (MC4100Z1) was documented previously [4]. T7 lysozyme's inhibitory effect on cell growth was documented in several previous studies [5-7]. We have added [5] as a reference in the revised manuscript.

e. Are there any other factors (such as nutrients), which could generate metabolic activity patterns?

We assume that the reviewer is referring to the spatial dependence of the gene expression capacity. Based on the literature, we suggest that the overall slower gene expression by cells in the interior of a microcolony can be caused by contact inhibition [1] and mechanical stress [2], in addition to lower local nutrient concentrations [8, 9].

2) What is the role of LuxR in the annular pattern? AHL activates LuxR in order to induce production of mCherry and T7 Lysozyme. This creates an AND gate like pattern between LuxR and AHL. If LuxR is not present in the cell, would there be an mCherry production? Is LuxR production linked to the metabolic state?

The reviewer is correct that LuxR and AHL forms an AND gate in driving the inhibition module. Without LuxR, we expect little or no mCherry expression. Also, LuxR expression should be linked to the metabolic state. However, for our system, AHL is the major limiting factor, because it is diluted throughout the entire droplet. In contrast, LuxR is confined in the cells. As such, *in comparison to AHL*, LuxR can be considered always ON in the full circuit.

As noted above, we have carried out additional measurements with MC4100Z1 cells carrying two "mutant" circuits. Both are the same as the full circuit except that one does not contain LuxR and the other does not contain LuxI. These experiments were done using the same condition we used to observe the ring patterns shown in Figs. 1 and S4. Neither mutant circuit generated detectable mCherry rings (Fig. S6). These experimental results directly demonstrate the essentiality of AHL signaling for the obtainment of the mCherry ring pattern. They also highlight that the patterns obtained in our study are indeed critically dependent on the synthetic gene circuit.

3) The 1D concentrations be related to physical units of molarity for comparison between experimental and computational results? The use of Δ ($=10 \mu\text{m}$) for the 1D spatial unit makes the paper more difficult to relate to physical parameters. Using μm in the figures and their legends would be make it more clear. Also could the 1D simulation be interpreted to be analogous to an average over all angles radiating from the center of the colony. Assigning a physical interpretation

The reason we did not assign molar units to computational results involving molecular concentrations is that, while we do state that one grid in the x-domain corresponds to $10 \mu\text{m}$, it is unclear that the x-, y-, and z-directions should all be $10 \mu\text{m}$. In fact, this is not the case in the experimental system. Thus, we kept concentrations to $\#/\Delta$ to reflect that.

4) A major discrepancy between the model and experiments is the mCherry signal at the center of colonies. While strong mCherry signals are observed at the center and the ring in the experiments, the model only accounts for the ring patterns. What causes this discrepancy?

There is actually no qualitative discrepancy. The apparent mCherry core is a ring-like pattern in the Z-direction. As shown in the confocal image (Fig. S9), there is indeed no mCherry

expression at the core of the 3D structure of the microcolony. Our model indeed captured the essence the patterning process (the formation of ring) and made predictions that were fully validated experimentally (Figs. 3, 4).

5) *The authors excluded single microcolonies per well for their analysis, because their dynamics were significantly different (page 18). However, the difference explained by the authors seems to support their theory. The model would predict larger rings that appear later would be caused by lower initial cell numbers (since lower initial cell density would take longer time to accumulate AHL).*

The reviewer's intuition is correct. This is indeed what we observed. For agar droplets containing only a single initial cell, we only observed rings for the next time point (taken at 48 hours), and such rings were much larger than the rings observed for two or more microcolonies per droplet. We have added representative images of single microcolonies per droplet at 36-hr and 48-hr time points to prove this point (Fig. S15). *As the author points out, these results are completely consistent with our model prediction and thus act as additional supporting evidence for our mechanism.*

How would the number of initial cells affect the dynamics?

As the reviewer states, an increase in the number of initial cells decreases the size of the mCherry ring.

The size of the ring should correlate with initial cell number. Is this relationship something the model would predict? If not, why?

Yes, if these different cells initiate separate microcolonies, we would predict that the ring size in each colony would, on average, decrease with the number of microcolonies. Experimentally, it is difficult to reproducibly control the number of initial cells in each droplet. Thus, we relied on dilution of the starting culture to give, on average, the appropriate number of cells per droplet. As a result, the number of cells that are captured in each droplet is an inherently stochastic process that is very difficult to control in a repeatable manner.

However, changing the initial number of cells at constant droplet volume is equivalent to changing the droplet volume at a constant initial number of cells. As shown in Fig. 3, the model prediction is fully consistent with the experimental data: It takes longer for AHL to accumulate at low cell/volume ratios, and that delay in reaching the AHL threshold results in a larger mCherry forming at a later time point.

6) *Another critical assumptions of the model is stated by the authors on page 6: "the cell's global gene expression decreases with increasing distance from the microcolony edge". Because this is the key for the pattern generation, strong experimental support should be present.*

The spatial dependence of gene expression capacity is consistent with our current understanding of bacterial physiology. Contact inhibition [1] and mechanical stress [2], which are likely to be more significant at the interior of microcolony, provide probable explanations for lower gene expression in those regions. Another possible contributing factor (but likely to a lesser degree) is that a cell's gene expression capacity is higher at the edge of the microcolony due to greater access to nutrients. This hypothesis is consistent with experimental evidence and the literature [8, 9]. As suggested by the reviewer, we have also carried out an additional experiment to directly demonstrate this point. As shown in Fig. S7, a microcolony of MC4100Z1 cells constitutively expressing mCherry indeed exhibits higher mCherry expression at the edge of the growing microcolony for a number of different time points.

7) *How stable is AHL in under the experimental condition used in this study? Is there any way to measure (or estimate) the actual concentration of AHL? The authors claimed "AHL concentration gradually decreases due to reduced T7 RNAP strength (Figure 2A) (page8)", but this is only true when AHL degrades. If AHL is very stable, termination of production does not decrease the concentration. Experimental support for the degradation of AHL should be present.*

AHL degrades through hydrolysis when a degrading enzyme is absent. Its half-life is highly dependent on the bacterial strain, growth conditions, and pH and is typically on the order of hours. In our experiments, the pH is 6.5, and we use an AHL degradation rate estimate of 5×10^{-3} 1/min, which is consistent with estimates provided in [10] based on experimental data and mathematical fitting. This AHL degradation rate is also within 3-fold of a published

measurement ([11]: 0.0018 1/min) and within 4-fold of published estimates ([12]: 0.0173 1/min; [13]: 0.0017 1/min) from other references under various experimental conditions.

8) *The brightness of mCherry signal on the ring does not look homogeneous. Some part of the ring looks brighter than the other part of the ring. How do the authors account for this observation?*

Indeed, the brightness of mCherry signal on the ring is not always homogeneous (as is evident in Fig. 1 and Fig. S4). We found that this heterogeneity is often caused by the location of the microcolonies in relation to the presence of neighboring colonies.

9) *What is the "dip" in the colony radius plot in fig1D? There seems to be a decrease in colony radius around 30hr. Is this "dip" real or some kind of artifact?*

The apparent "dip" is due to the variability in data processing when applying our image-analysis algorithm to the time-lapse images, though the algorithm overall is accurately depicting the microcolony's edge on average.

10) *An interesting prediction of the model is the two phase growth behavior during colony expansion. The colony radius initially increases rapidly, and increases again after slow expansion period (Figure 1F). Does this agree with the experimental results? If so, how does this fit in the size control mechanism the authors discussed in page 11?*

We are glad that the reviewer finds this prediction interesting. Indeed, the model prediction depicted in Fig. 1F agrees with the experimental result (Fig. 1D). We believe that this size-control mechanism is analogous to the chalone hypothesis. In the chalone hypothesis, the "chalone" is defined as a secreted, diffusible growth inhibitor. Briefly, the chalone accumulates as an organ or tissue grows in size. Once the chalone reaches a threshold concentration in an organism or a specific organismal compartment, enough growth inhibition occurs such that it results in a stoppage of growth of the organ or tissue. This organ or tissue size is then fixed from that point forward. In our system, the equivalent of the chalone is AHL. Once AHL reaches a threshold concentration, it will result in an extended stoppage of growth of the microcolony through the activation of lysozyme.

11) *The two rings in Fig4c are not very clear to me. Better representative images should be shown. Also, why does the 1st ring seem to be shifted to the inside from 36hr (around 700um) to 48hr (500um)?*

We do have other representative double-ring patterns (an additional representative image of one is shown in Fig. S14), but Fig. 4C presents the clearest double-ring pattern. Regarding the reviewer's question, we did try to capture the growth region as best as possible by taking fluorescent images. As the microcolony expanded at the 48-hr time point, this growth region became larger. We thus captured a different frame for the 48-hr time point than we did from the 36-hr time point. Because the frames were different, we chose a different sector with which to quantify the mCherry signal for varying radii. This somewhat different sampling likely resulted in a minor shift in where the peak seemed to occur. The fluctuations associated with stochastic gene expression as well as noise in the image capturing and image analysis could also have contributed to this minor shift.

12) *Adding control experiments with the strain which only carries an activation or inhibition module would make enhance the strength of the story.*

We thank the reviewer for the suggestion. As noted in responses above, we have now included additional control experiments, which reinforced several points:

First, the experiment involving MC4100Z1 carrying a single reporter illustrated the spatial-dependent gene expression capacity within a growing microcolony (Fig S7).

Second, we have carried out additional measurements with MC4100Z1 cells containing two "mutant" circuits. Both are the same as the full circuit except that one does not contain LuxR and the other does not contain LuxI. These mutant circuits are more appropriate than single-component gene circuits since their resulting fluorescent spatial distributions can shed light on the direct role of AHL signaling in the pattern formation process. These experiments were done using the same condition we used to observe the ring patterns shown in Figs. 1 and S4. Neither mutant circuit generated detectable mCherry rings (Fig. S6). These experimental results directly demonstrate the essentiality of AHL signaling for the obtainment of the mCherry ring pattern.

They also highlight that the patterns obtained in our study are indeed critically dependent on the synthetic gene circuit.

Again, we would like to reiterate the central point of our analysis: *In our experimental system with the full synthetic gene circuit, the morphogen (AHL) serves as a timing cue, instead of a spatial cue, to trigger the formation and maintenance of the ring patterns and to modulate the ring size.* In addition to the new control experiments, this central point has been rigorously validated in our original submission through experiments perturbing the full circuit (Figs 3-4) and the corresponding modeling analysis. In these experiments, the predicted role of AHL is fully validated by experiments for varying domain size and initial AHL concentration.

Major Presentation Point

The x-axis representing time for Figure 1f should be extended to match Figure 1d for comparison between computational and experimental results.

We should clarify that the goal of our model was to capture the essential, qualitative features of the system dynamics. These include the generation of a ring with constant size over time, the dependence of the ring size on initial AHL concentration and domain size, and the ability to generate double rings. The model was indeed able to capture all of these features. However, due to drastic simplification of the model, in terms of both cellular dynamics and spatial configuration, we do not expect an exact match between modeling and experiments in quantitative aspects.

Minor Presentation Points

The figure quality needs to be improved for publication. Specifically:

1) The solid lines red and blue lines are too thick causing the following issues:

We have decreased the thickness of the red and blue lines accordingly in the revised manuscript.

i. In Figure 1c obscure the primary data indicated by the blue and green dots that appear to be actually dashed lines. If the dots are actually dashed lines, the figure legend should be corrected.

The cyan and green dots are indeed dots and not dashed lines. Perhaps this will be clearer with the thinner red and blue lines.

ii. It is difficult to see variation of the mCherry ring location in 1d

iii. It is difficult to see what appears to be noise of the red and green lines in 1b

We have decreased the thickness of the red line in 1D and of the red and blue lines in 1C accordingly in the revised manuscript. Perhaps both of these will be clearer now.

2) The location of the fluorescent reporters in the circuit not very clear from Figure 1a

We have edited the Fig. 1A figure legend accordingly to illustrate the location of the fluorescent reporters in the revised manuscript.

3) Figure 1b should have a color bar for better quantitative representation and indication of the saturation and would be clearer if an additional visual aide were given for the time progression from left to right.

We have added a color bar to Fig. 1B to better indicate saturation and background level. We have increased the font size of the white numbering in the phase images corresponding to the time point at which the image was taken to increase the clarity of the time progression from left to right. We have also added a large arrow horizontally below Fig. 1B to better indicate that time progression is from left to right.

The relationship of the coloring to Figure 1 could be made more clear.

We had difficulty interpreting this comment.

4) Inclusion of the y-axis label in Figures 2a-d would be helpful. The split y-axis between the three time points in panels b and c make quantitative comparison difficult. Figure 2d is particularly not clear without axis labels. For the AND gate, more explicit labeling of the gate would be helpful.

Changes were made to Fig. 2 accordingly in the revised submission.

5) Comparison of the three magenta traces in Figure 3a is difficult.

The three magenta traces were made thicker for better comparison in the revised submission.

Minor Points

1) The strain MC4100Z1 should be described in more detail or a reference should be given with such a description.

We have added a description for MC4100Z1 in the revised manuscript.

Reviewer #3 :

In their manuscript "Temporal control of self-organized pattern formation without morphogen gradients in bacteria," Payne et al. report a synthetic gene circuit that generates a ring pattern in the apparent absence of a morphogen gradient. Synthetic gene circuits that give rise to similar patterns have been demonstrated previously, but they have relied on spatial inhomogeneity of intercellular signaling molecules to drive pattern formation.

In contrast, the authors argue that their circuit generates a stable pattern based on the temporal dynamics of a spatially homogeneous intercellular signal (AHL). Combined with a spatial inhomogeneity in overall gene expression and cell growth, a global increase in AHL initiates a positive-feedback loop that maintains the pattern even as the microcolony continues to grow.

The authors argue convincingly that their system's patterns form independently of a morphogen gradient. Their agent-based model successfully recapitulates the major features of the system, predicting several behaviors that are experimentally verifiable. The work is exciting and has significance both for efforts to understand natural pattern formation and for engineering them in synthetic systems, and I think it deserves publication.

We thank the reviewer for recognizing the novelty and significance of our work.

However, there are several points that could make their argument stronger and easier to follow.

Major comments:

1. *The initial experimental evidence that pattern formation proceeds in the absence of an AHL gradient rests on the exogenous application of 100 nM AHL; however, not enough context is given to evaluate the claim that this effectively abolishes any extant AHL gradient. While adding AHL obviously changes the relative strength of a hypothetical spatial gradient, the authors in fact still saw an effect: a smaller pattern. I don't think that additional experiments are called for, but putting this experiment in better context would make it more convincing. For example, is there a reference that could be cited where an AHL gradient does modulate a spatial effect, and where an additional 100 nM abolishes that effect?*

We thank the reviewer for the insightful comment. We have revised the text to further clarify this point, which is based on our current understanding of the kinetic properties of the LuxR/LuxI/AHL system.

First, we note that 100 nM of AHL is approximately the saturating concentration for full induction of the lux promoter (when sufficient LuxR is present), which has a half-activation concentration of ~10-30nM (e.g., see [14]). Based on our experience, even if LuxI is constitutively expressed from a strong promoter (e.g, lac promoter), the "quorum" required to generate sufficient AHL (10~30nM) to trigger the *luxI* promoter is ~10⁶-10⁸ bacteria/ml (depending on experimental conditions), or 5,000-500,000 bacteria per droplet (5 μl). Since our experiment starts from ~1-4 cells and there is a delay in LuxI expression (controlled by T7RNAP), we expect the initial AHL concentration (at any spatial location) produced by the bacteria to be orders of magnitude lower than 100 nM. As such, the exogenous addition of 100 nM AHL should indeed abolish any initial AHL gradient.

2. *One of the strengths of model-based circuit characterization is in identifying non-obvious properties of the system that are crucial for producing the observed (or desired) behavior. This is exactly what happened when the authors observed that spatially dependent variation in global gene expression was essential for their model to recapitulate the observed phenomena. It's not clear, however, whether "increased global gene expression", "faster cell growth" and "high metabolic capacity" at the edge of the growing microcolony refer to the same phenomenon, or different but coupled phenomena (cf. Scott et al, Science 330:1099 (2010)). A better explanation of the relationship between these terms (concepts? phenomena?) would strengthen the argument for including a seemingly arbitrary model component.*

We are glad that the reviewer noted the value of modeling in defining a previously under-appreciated factor that is critical for pattern formation: namely, the spatial-dependent gene expression capacity in a microcolony. We apologize for the lack of consistency in the use of several related terms, as noted by the reviewer. We have revised the manuscript to ensure a consistent and clear description of these terms. Furthermore, we have provided additional experimental evidence for this assumption. Briefly, the cell strain of the study (MC4100Z1) was transformed with an aTc-inducible reporter (mCherry). Upon full induction, this strain exhibited higher mCherry expression on the edge of the microcolony for several different time points, directly demonstrating the spatial-dependent gene expression capacity (Fig. S7).

3. Finally, the authors propose that the location of lysozyme (and RFP) expression is determined by what amounts to a logical AND gate between AHL level and the cells' metabolic capacity (Figure 2D). This conceptual model is neat, logical and accounts for the behavior they observe; but it's not clear how well their results support it. Does varying the AND gate's other input (metabolic capacity), either in simulation and/or experiment, also modulate the pattern size?

Experimentally, it would be extremely difficult, if not impossible, to vary metabolic capacity without changing such parameters as growth rate, growth capacity, etc. However, we have provided simulation data where we have added a multiplicative factor to ϕ (which we have called α) and perturbed it accordingly (Fig. S16). The effect of higher α values is a corresponding decreasing in ring radius. This effect is intuitive since higher α values lead to higher AHL (and thus lysozyme) synthesis at the edge of the microcolony. This leads to a decrease in the time necessary to reach the AHL threshold necessary to trigger ring formation, and therefore, a smaller ring forms for higher α values.

Alternately, is there a direct measure of metabolic capacity (perhaps an integrated, constitutively expressed fluorescent marker) that could be used to bolster the assertion that metabolic capacity is higher on the outer edge of the growing microcolony?

Yes, the data of this very experiment has been added in Fig. S7. Note that fluorescence is higher on the edge of such a microcolony for a number of different time points, proving our assumption that gene expression is higher on the edge of the growing microcolony.

Minor corrections:

1. Is a multi-well coverslip really a "microchannel device"? To my mind, "microchannel" implies that something is flowing through it.

We have changed our terminology in the revised manuscript such that "microchannel device" is now referred to as a "multi-well device."

2. I found the Y-axis labels in Figure 2 to be terribly confusing: the measurement being indicated appears to be at the top of the plots (or nowhere at all, as in D), and there are no units (except in C). Also, in the stacked plots (B and C), only a single tic label is given for each plot, so it's not immediately obvious why there's a "70" or a "3" floating next to the plot. Please either label the Y axes more completely; or, if the units are unimportant, indicate that the scale is arbitrary and omit them entirely.

We have fixed the y-axes accordingly in the revised manuscript.

3. I found your timelapse movies quite striking. If you have one for the double-ring experiment (and there's room left in your SI allocation), you might consider including it.

We are glad that the reviewer appreciates the time-lapse movies provided. Unfortunately, we do not have a time-lapse movie of the double-ring experiment. However, the successive images from the experiment at the 36-hr and 48-hr time points are sufficient to prove that the two rings emerge sequentially, which is the essential point we are attempting to make in Fig. 4. We have also provided an image of another representative double-ring pattern (Fig. S14).

4. For Figures S2 and S3, you might consider moving the experimental protocols from the figure legends into a separate block of SI text.

We have made the appropriate stylistic changes in the revised manuscript.

5. For the experiment described in Figure S2 and the accompanying legend, please describe pLysS. It was not clear until I looked it up that it expresses lysozyme under control of a lac operator.

Actually, in pLysS, the T7 lysozyme coding sequence is in the antisense orientation relative to the *ptet* promoter, such that only a small amount of T7 lysozyme is expressed. The gene is not under the control of a *lac* operator. Thus, in this experiment, IPTG is added to tune T7 RNAP expression from the activation module, which is under the control of a *lac* operator. In any case, the point is well taken, and we have described the pLysS plasmid in detail in the revised manuscript. We thank the reviewer for the suggestion. In addition, the original reference for the plasmid [5] is provided in the revised submission.

Reviewer #4:

In this manuscript, the authors tried to demonstrate that the self-organized pattern formation could be achieved independent of the morphogen gradient. They employed an engineered E. coli strain that carrying two plasmids (circuits), an activator and an inhibitor, as an experimental model for pattern formation analysis. They presented that the ring patterns generated by the E. coli colonies (reported by mCherry) formed in soft agar were not determined by a spatial cue due to the morphogen gradient, but were self-regulated in a timing cue. The authors concluded that a novel mechanism on pattern formation was demonstrated, according to their experimental evidence along with the theoretical simulation. The idea/model on pattern formation mechanism is nice and highly appreciated, but I do feel regret that the results presented in this study could not persuade me to fully accept their conclusion.

We thank the reviewer for appreciating the central idea of our pattern formation mechanism and for specific suggestions to improve the rigor of our conclusion. We hope that our clarifications and additional results will address the technical concerns the reviewer has.

Briefly, two major points seriously prevented me to support the authors' statement.

1. It is doubted whether the results fully supported the conclusion. It is true that AHL was the only controlled morphogen in the study, but it didn't mean only a single signal/morphogen determine the pattern formation. Other elements, which were not detected or evaluated in this study, for instance, those involved in the so-called metabolic burden, possibly contributed to the ring pattern.

The reviewer is absolutely correct that AHL is not the only signal that determines the pattern formation process. In addition, we agree that other elements, particularly the metabolic state of cells, also play a role. This is precisely the point we attempted to establish in interpreting our data. For example, as illustrated in Figure 2, the initiation of the mCherry ring depends on both a sufficiently high AHL concentration (temporal cue) and a sufficiently high gene expression capacity.

2. It seems that only several colonies were selected for data processing.

Microcolonies were excluded if they were too close in proximity to the edge or another microcolony or if there was only one microcolony per well. In the former case, it was assumed that such microcolonies would be 1) too unreliable to quantitate given that only a very small sector would be available for processing and 2) unrepresentative of microcolonies in isolation that are free to expand spatially for several hundred μm s. In the latter case, a single microcolony would form a larger ring later than other microcolonies (see Fig. S15). *Note that this is completely consistent with our proposed mechanism and mathematical modeling since a lower cell to volume ratio causes AHL's path to the threshold concentration to take longer, resulting in a larger mCherry ring forming later.*

I'm wondering if such a few data sets are sufficient for estimation. It's hard to say whether the observation on ring formation was statistically reliable.

Although the microcolonies discussed above were excluded, the other microcolonies were all chosen for data analysis. The formation of the ring was highly reproducible (Figs. 1, 3, and S4). When comparisons of ring sizes were made, we also had sufficient reliable data to make statistically significant analysis. As is evident in Figs. 3C and 3E, the error bars (each calculated from sets of 7, 10, 9, and 5 and 3, 6, and 6 replicate colonies, respectively) are very small relative to the interval over which the trends span, indicating that the trend is indeed statistically significant.

Also, no negative controls were provided, for instance, the controls of the cells carrying a single circuit, and/or the cells carrying the reversed genetic constructions, etc.

We have provided new experimental evidence with the same cell strain (MC4100Z1) containing a plasmid encoding constitutively expressed fluorescence (mCherry). The results in Fig. S7 (discussed above) support our assumption of spatial-dependent gene expression critical to the mechanism and mathematical modeling.

It is unclear what the reviewer referred to by “the reversed genetic constructions.” Please refer to our response to reviewer #2’s point 12 for an explanation of why we believe that experiments with the activation module only or inhibition module only are not necessarily appropriate for this particular study and a description of additional controls with LuxR and LuxI knockouts (the results of which are displayed in Fig. S6).

I'm afraid the so-called "robust" (repeated in the manuscript) pattern formation does not sound so much "robust".

Robustness is defined as the persistence of a system’s characteristic behavior. We believe that the robustness of the pattern formation is aptly demonstrated in Fig. S4, where we show similar patterns arising for six independent clones, captured during 5 different, independent experiments. These patterns arise despite variations in experimental conditions, including droplet composition, agar density, and number of cells per droplet. Additional experiments with the same gene circuit in a different cell strain using a different experimental setup revealed a similar mCherry ring pattern (Fig. S5). Therefore, we consider the pattern formation arising from the gene circuit’s dynamics to be robust.

Minor comments:

1. The authors used the phrase "robust, self-organized pattern" repeatedly in the manuscript. What does the terms "robust" and "self-organize" precisely refer to?

Robustness is defined as the persistence of a system’s characteristic behavior. We have defined it in the latest revision of the manuscript. Self-organized is defined as being independent of pre-defined spatial cues. We have added this definition to the revised manuscript as well. A classic example of a system that is not self-organized is one that responds to an external spatial morphogen gradient (i.e., receiver cells responding to sender cells constituting a separate system). *We call our system self-organized since all of the components critical to forming a pattern (AHL, gene regulatory network, etc.) are embedded in a single isogenic cell population and do not require a pre-defined spatial cue.*

2. p3, 1st paragraph, the last sentence: "...the generation of robust, self-organized pattern of gene expression in the absence of an apparent morphogen gradient" was confusing. It needs to be fixed.

We have changed the original sentence from “However, using *E. coli* programmed by a synthetic gene circuit, we demonstrate here the generation of robust, self-organized patterns of gene expression in the absence of an apparent morphogen gradient.” to “However, using *E. coli* programmed by a synthetic gene circuit, we demonstrate here the formation of self-organized patterns without an apparent morphogen gradient.” We hope that this sentence is now clearer.

3. Why a special device of microchips was used for the colony formation? Is it advantageous in this study? Is it possible that the ring pattern of the colonies was somehow device-dependent?

We would like to reiterate the central conclusion of our study: *In this synthetic pattern-forming system, the morphogen (AHL) serves as a timing cue, instead of a spatial cue, to trigger the formation and maintenance of the ring patterns.* With a larger, standard device (i.e., an 8-cm plate), there would be a spatial gradient of AHL. Thus, an examination of the system would need to account for this spatial gradient, which would complicate our goal of characterizing the influence of AHL’s temporal dynamics on the pattern formation process.

The ring pattern is indeed device-dependent. Most critically, the volume of the device directly influences the size of the ring pattern (Fig. 3E). Specifically, the larger the domain size (or volume), the longer it takes for AHL to accumulate to the threshold concentration necessary to trigger ring formation. This delay in the timing of ring initiation results in a larger pattern.

That being said, we were able to demonstrate ring formation using the same synthetic gene circuit in a different cell strain (MG1655) under a different experimental condition (on a rigid agar surface) as is demonstrated in Fig. S5, suggesting that ring pattern formation is not restricted to the specific device that is at the center of this study.

4. *Why 1-4 cells were inoculated in each well/chamber? Are there any reasons in particular? Would the initial numbers of cells play a role in the ring formation?*

We have diluted to only a small number of cells per well such that each growing microcolony is not interfered with by neighboring microcolonies in close proximity. In this manner, microcolonies can have enough room to expand and give rise to ring patterns before growth from another microcolony can collide with it.

Yes, we do expect the number of cells to play a role in the ring formation process. Namely, if these different initial cells initiate separate microcolonies, we would predict that the ring size in each colony would, on average, decrease with an increasing number of microcolonies. Experimentally, it is difficult to reproducibly control the number of initial cells in each droplet. Thus, we relied on the proper dilution of cells to give, on average, the appropriate number of cells per droplet. As a result, the number of cells that are captured in each agar droplet is an inherently stochastic process that is very difficult to control in a repeatable manner.

However, changing the initial number of cells at a constant droplet volume is equivalent to changing the droplet volume at a constant initial number of cells. As shown in Fig. 3, the model prediction is fully consistent with the experimental data: namely, that it takes longer for AHL to accumulate at low cell/volume ratios, and that delay in reaching the AHL threshold results in a larger mCherry ring forming at a later time point.

5. *The figure legends contained insufficient information. For example, what the "#" in Figs. 2 and 4 stands for? I'm not sure if I caught the proper meaning of the figures.*

We apologize for the confusion. “#” indicates the number of molecules. We have revised the figure legends for clarification.

6. *The authors are recommended to pay attention to the formats in the manuscript, such as, the inconsistent formats of references, the differences in the calligraphy between the figures (small letters) and the figure legends (capital letters).*

We have made such revisions to the formatting in the latest revised manuscript.

1. Morse, R.P., et al., *Structural basis of toxicity and immunity in contact-dependent growth inhibition (CDI) systems*. Proc Natl Acad Sci U S A, 2012. **109**(52): p. 21480-5.
2. Jozefczuk, S., et al., *Metabolomic and transcriptomic stress response of Escherichia coli*. Mol Syst Biol, 2010. **6**: p. 364.
3. Borello, U. and A. Pierani, *Patterning the cerebral cortex: traveling with morphogens*. Curr Opin Genet Dev, 2010. **20**(4): p. 408-15.
4. Tan, C., P. Marguet, and L. You, *Emergent bistability by a growth-modulating positive feedback circuit*. Nat Chem Biol, 2009. **5**(11): p. 842-8.
5. Studier, F.W., *Use of bacteriophage T7 lysozyme to improve an inducible T7 expression system*. J Mol Biol, 1991. **219**(1): p. 37-44.
6. Dubendorff, J.W. and F.W. Studier, *Creation of a T7 autogene. Cloning and expression of the gene for bacteriophage T7 RNA polymerase under control of its cognate promoter*. J Mol Biol, 1991. **219**(1): p. 61-8.
7. Dubendorff, J.W. and F.W. Studier, *Controlling basal expression in an inducible T7 expression system by blocking the target T7 promoter with lac repressor*. J Mol Biol, 1991. **219**(1): p. 45-59.
8. Klumpp, S., Z. Zhang, and T. Hwa, *Growth rate-dependent global effects on gene expression in bacteria*. Cell, 2009. **139**(7): p. 1366-75.
9. Scott, M., et al., *Interdependence of cell growth and gene expression: origins and consequences*. Science, 2010. **330**(6007): p. 1099-102.
10. You, L., et al., *Programmed population control by cell-cell communication and regulated killing*. Nature, 2004. **428**(6985): p. 868-71.
11. Kaufmann, G.F., et al., *Revisiting quorum sensing: Discovery of additional chemical and biological functions for 3-oxo-N-acylhomoserine lactones*. Proc Natl Acad Sci U S A, 2005. **102**(2): p. 309-14.
12. Liu, C., et al., *Sequential establishment of stripe patterns in an expanding cell population*. Science, 2011. **334**(6053): p. 238-41.

13. Song, H., et al., *Spatiotemporal modulation of biodiversity in a synthetic chemical-mediated ecosystem*. Nat Chem Biol, 2009. **5**(12): p. 929-35.
14. Collins, C.H., J.R. Leadbetter, and F.H. Arnold, *Dual selection enhances the signaling specificity of a variant of the quorum-sensing transcriptional activator LuxR*. Nat Biotechnol, 2006. **24**(6): p. 708-12.

2nd Editorial Decision

23 August 2013

Thank you again for submitting your work to Molecular Systems Biology. We have now heard back from the three referees who agreed to evaluate your manuscript. As you will see from the reports below, while the main concerns of reviewers #3 and #4 have been satisfactorily addressed, reviewer #1 has requested additional experimentation to demonstrate the role of metabolic heterogeneity in pattern formation.

We have contacted reviewer #3 regarding this issue and the response was the following: "Reviewer #1 correctly points out that fluorescent protein maturation depends on oxygen availability, and that another possible explanation of an advancing mCherry ring is the presence of an oxygen gradient along the radius of the growing *E. coli* microcolony. However, given the spatial scale under consideration (< 1 mm) and oxygen's diffusivity through agar (about 10^{-3} mm²/s), it strikes me as unlikely that oxygen deprivation is the limiting fluorescent protein maturation. To my eye, Reviewer #1's insistence on modulating global gene expression levels by changing the growth conditions (in particular by changing the carbon source) seems to be misplaced. It is true that one might reasonably expect slower overall growth leading to lower overall gene expression levels -- but, as the authors point out in their rebuttal, the critical point is the existence of spatial heterogeneity in gene expression, not the precise mechanism by which it arises. It's not clear what conclusions could be drawn from Reviewer #1's proposed experiments because it's not clear *a priori* how a global change in gene expression would be expected to affect its spatial heterogeneity in a growing microcolony. In the end, a direct measure of gene expression is required and, given the constraints of their experimental setup, the authors' choice of a fluorescent protein reporter seems appropriate. Their results (an advancing mCherry ring at the edge of the microcolony) are consistent with their proposed model and further substantiate their main claim, that rising global AHL levels serve as a timing signal to give rise to spatial phenotypic heterogeneity." In line with the additional comments provided by reviewer #3, we feel that this additional experiment would not be required. However, reviewers #1 and #3 have raised a number of minor points that we would ask you to address in a revision of the manuscript.

Referee reports:

Reviewer #1:

Direct evidence to demonstrate the existence and the role of metabolic heterogeneity in pattern formation is still lacking. The proposed demonstration (Fig S7) is not sufficient. Fluorescent proteins are problematic reporters of the cell's metabolic state insofar as they rely on high intracellular oxygen to fluoresce (Heim et al. (1994) Wavelength mutations and posttranslational autoxidation of green fluorescent protein. Proceedings of the National Academy of Sciences USA 91: 12501). An equally-plausible interpretation of the mCherry experiment is that there is an oxygen gradient through the microcolony (which is true). How that oxygen gradient can be de-convolved from metabolic state remains to be demonstrated.

The control experiments suggested previously, reproduced below:

'There is an established quantitative relationship between nominal nutrient quality and gene expression levels (Klumpp et al., 2009) - to substantiate the proposed mechanism (and the model

parameterization), the experiments in Fig 1B should be repeated with a variety of nutrient agar (in addition to the LB supporting about 2.5 dbl/h). A good range would be 0.4% glucose minimal MOPS media (supporting about 1 dbl/h) and 60 mM acetate minimal MOPS media (supporting about 0.3 dbl/h). That should change the basal k_T , k_L and k_A synthesis rates according to Klumpp. Without this additional validation, the proposed mechanism is unsubstantiated.'

appear to have been misinterpreted. All nutrients in these media are in saturating amounts. It is not a nutrient gradient that one needs to establish, but rather a global change in the growth rate of all cells in the colony. One way to do this is by changing the carbon source.

Demonstration of a predictable change in the ring structure with an absolute shift in the proliferation rate of the cells in the colony would provide strong indirect evidence in support of the authors' claimed metabolic gradient. Beyond being a critical control for the proposed mechanism and model, if the circuit only works in LB or 2xYT, then the results are of limited interest.

Further, on page 7 of the revised manuscript, the statement:

'Another possible contributing factor is that a cell's gene expression capacity increases with local nutrient concentration (Klumpp et al, 2009; Scott et al, 2010).'

is not correct. The cited references do not claim any correspondence between gene expression and nutrient concentration - in exponential growth, all nutrients are present in saturating amount; it is the chemical composition of the substrates that is important. E coli catabolizes glucose more rapidly than glycerol, and can sustain a faster growth rate with glucose as a carbon source. The concentration of the carbon source is not varied in these studies.

Reviewer #3:

All of my concerns were adequately addressed, with one minor exception: I asked if the application of 100 nM exogenous AHL was sufficient to abolish an AHL gradient, and the authors explained in their rebuttal that 100 nM AHL was enough to saturate LuxR. That's sufficient explanation for me -- but it wasn't included in the manuscript. Please do!

Recommended for publication.

Reviewer #4:

The revised manuscript was improved by supplying with a long supplementary information. I do appreciate that the authors performed a number of additional experiments to address the technical concerns, in particular, the highly essential experiments shown in the Figures S6 and S7, which supported the ring formation in evidence.

2nd Revision - authors' response

02 September 2013

We thank the reviewers for evaluating our work. We are glad that Reviewers #3 and #4 were satisfied with our previous revision. We hope that this next revision, in combination with our responses below, will address the remaining concerns of Reviewer #1.

With these new clarifications and corresponding revisions, we hope you and the reviewers will find the manuscript sufficiently improved and acceptable for publication.

Point-by-point responses to editorial and reviewer comments

Editorial comments

Thank you again for submitting your work to Molecular Systems Biology. We have now heard back from the three referees who agreed to evaluate your manuscript. As you will see from the reports below, while the main concerns of reviewers #3 and #4 have been satisfactorily addressed, reviewer #1 has requested additional experimentation to demonstrate the role of metabolic heterogeneity in pattern formation.

*We have contacted reviewer #3 regarding this issue and the response was the following: "Reviewer #1 correctly points out that fluorescent protein maturation depends on oxygen availability, and that another possible explanation of an advancing mCherry ring is the presence of an oxygen gradient along the radius of the growing E. coli microcolony. However, given the spatial scale under consideration (< 1 mm) and oxygen's diffusivity through agar (about 10^{-3} mm²/s), it strikes me as unlikely that oxygen deprivation is the limiting fluorescent protein maturation. To my eye, Reviewer #1's insistence on modulating global gene expression levels by changing the growth conditions (in particular by changing the carbon source) seems to be misplaced. It is true that one might reasonably expect slower overall growth leading to lower overall gene expression levels -- but, as the authors point out in their rebuttal, the critical point is the existence of spatial heterogeneity in gene expression, not the precise mechanism by which it arises. It's not clear what conclusions could be drawn from Reviewer #1's proposed experiments because it's not clear *a priori* how a global change in gene expression would be expected to affect its spatial heterogeneity in a growing microcolony. In the end, a direct measure of gene expression is required and, given the constraints of their experimental setup, the authors' choice of a fluorescent protein reporter seems appropriate. Their results (an advancing mCherry ring at the edge of the microcolony) are consistent with their proposed model and further substantiate their main claim, that rising global AHL levels serve as a timing signal to give rise to spatial phenotypic heterogeneity." In line with the additional comments provided by reviewer #3, we feel that this additional experiment would not be required.*

However, reviewers #1 and #3 have raised a number of minor points that we would ask you to address in a revision of the manuscript.

We will make the minor revisions accordingly and explain them below.

On a more editorial level, I would like to draw your attention to the following:

- The quality of the equations seems rather poor, so we would like to ask you to re-insert them and make sure they display well.*
- Please provide individual figure files for the main figures.*
- Part of the text in the Supplementary Information is in red; this needs to be changed.*

We have made the appropriate changes in the latest revision.

Reviewer #1:

Direct evidence to demonstrate the existence and the role of metabolic heterogeneity in pattern formation is still lacking. The proposed demonstration (Fig S7) is not sufficient. Fluorescent proteins are problematic reporters of the cell's metabolic state insofar as they rely on high intracellular oxygen to fluoresce (Heim et al. (1994) Wavelength mutations and posttranslational autoxidation of green fluorescent protein. Proceedings of the National Academy of Sciences USA 91: 12501). An equally-plausible interpretation of the mCherry experiment is that there is an oxygen gradient through the microcolony (which is true). How that oxygen gradient can be de-convolved from metabolic state remains to be demonstrated.

We agree with the reviewer that we cannot exclude the possibility of an oxygen gradient within the microcolony. However, as we stated in our previous correspondence, “*Regardless of the underlying molecular mechanism, the critical issue is whether cells at the edge [of the microcolony] are able to express genes at a faster rate.*” In our first round of revision, we have provided experimental evidence that directly demonstrated this point (Fig. S7).

We hypothesized previously that spatially heterogeneous mechanical stress, cell-cell contact inhibition, and nutrient consumption can lead to such an effect. An oxygen gradient also presents one plausible explanation of spatial-dependent gene expression throughout the microcolony. In light of the reviewer’s comments, we have now added a sentence, along with an appropriate reference, in the revised manuscript to reflect that possibility. However, the physiological mechanism by which spatial-dependent gene expression occurs is not the subject of this study.

In addition, we note that Fig. S7 indeed demonstrates the existence of metabolic heterogeneity. The reviewer is correct that fluorescent proteins rely on sufficiently high intracellular oxygen levels to fluoresce. However, if oxygen were indeed the limiting factor influencing fluorescence levels, the fluorescence intensity should decrease over time as the microcolony expands. This was not the case. To avoid a saturating fluorescence signal when imaging the colony shown in Fig. S7a, we exposed the cells to light at a wavelength of 546 nm for 10 s, 3.5 s, and 1.25 s for the 18-hr, 24-hr, and 30-hr time points, respectively. For the colony in Fig. S7b, we had to expose the cells to light of the same wavelength for 6 s, 3 s, and 0.65 s for the 18-hr, 24-hr, and 30-hr time points, respectively, to avoid saturation. Since the fluorescence intensity increased over time, the fluorescent patterns were highly unlikely to result from oxygen limitation. Also, in the critical experiments displayed in the main figures, mCherry fluorescence intensity increases significantly after ~20-50 of hours before any decrease in signal intensity occurs. Therefore, the fluorescent reporters are indeed accurately reflecting the qualitative dynamics of the synthetic gene circuits in our study.

The control experiments suggested previously, reproduced below:

‘There is an established quantitative relationship between nominal nutrient quality and gene expression levels (Klumpp et al., 2009) - to substantiate the proposed mechanism (and the model parameterization), the experiments in Fig 1B should be repeated with a variety of nutrient agar (in addition to the LB supporting about 2.5 dbl/h). A good range would be 0.4% glucose minimal MOPS media (supporting about 1 dbl/h) and 60 mM acetate minimal MOPS media (supporting about 0.3 dbl/h). That should change the basal k_T , k_L and k_A synthesis rates according to Klumpp. Without this additional validation, the proposed mechanism is unsubstantiated,’

appear to have been misinterpreted. All nutrients in these media are in saturating amounts. It is not a nutrient gradient that one needs to establish, but rather a global change in the growth rate of all cells in the colony. One way to do this is by changing the carbon source.

Demonstration of a predictable change in the ring structure with an absolute shift in the proliferation rate of the cells in the colony would provide strong indirect evidence in support of the authors' claimed metabolic gradient. Beyond being a critical control for the proposed mechanism and model, if the circuit only works in LB or 2xYT, then the results are of limited interest.

We appreciate the suggestion, and we agree that these additional experiments could test whether the same mechanism could be maintained under different growth conditions.

However, we would like to reiterate the central point of our analysis: *In our experimental system with the full synthetic gene circuit, the morphogen (AHL) serves as a timing cue, instead of a spatial cue, to trigger the formation and maintenance of the ring patterns and to modulate the ring size.* To this end, our experimental system, including the synthetic gene circuit and the corresponding growth conditions, is intended to serve as a *model system* to test this novel mechanism of pattern formation. While we have not tested the circuit dynamics using a

different growth medium, we have demonstrated qualitatively the same dynamics (self-organized ring pattern formation) in a different bacterial strain (MG1655) and in a different growth condition (an inkjet-printed colony on top of the agar surface). These results (Fig. S5) as well as the many replicate colonies showing similar behavior (Fig. S4) provide evidence for the robustness of the circuit function, as detailed in our previous revision.

Our central point has indeed been rigorously validated in our previous experiments perturbing the full circuit (Figs. 3-4) and the corresponding modeling analysis. In these experiments, the predicted role of AHL is fully validated by experiments for varying domain size and initial AHL concentration. In addition, two critical aspects of the mechanism (the essentiality of AHL signaling in the ring formation mechanism and the existence of spatial-dependent gene expression) have been aptly demonstrated in Figs. S6 and S7, respectively.

Further, on page 7 of the revised manuscript, the statement:

'Another possible contributing factor is that a cell's gene expression capacity increases with local nutrient concentration (Klumpp et al, 2009; Scott et al, 2010).'

is not correct. The cited references do not claim any correspondence between gene expression and nutrient concentration - in exponential growth, all nutrients are present in saturating amount; it is the chemical composition of the substrates that is important. E coli catabolizes glucose more rapidly than glycerol, and can sustain a faster growth rate with glucose as a carbon source. The concentration of the carbon source is not varied in these studies.

We thank the reviewer for the advice. We have reworded the above statement in the following manner to be more precise: "Another possible contributing factor is that a cell's gene expression capacity increases with the availability of growth-limiting chemical substrates."

Reviewer #3:

All of my concerns were adequately addressed, with one minor exception: I asked if the application of 100 nM exogenous AHL was sufficient to abolish an AHL gradient, and the authors explained in their rebuttal that 100 nM AHL was enough to saturate LuxR. That's sufficient explanation for me -- but it wasn't included in the manuscript. Please do!

We have included a sentence which states that 100 nM AHL is enough to saturate LuxR, along with an appropriate reference.

Recommended for publication.

We thank the reviewer for his or her constructive suggestions and for recommending our work for publication.

Reviewer #4:

The revised manuscript was improved by supplying with a long supplementary information. I do appreciate that the authors performed a number of additional experiments to address the technical concerns, in particular, the highly essential experiments shown in the Figures S6 and S7, which supported the ring formation in evidence.

We thank the reviewer for his or her constructive suggestions and for appreciating our diligence in providing additional experiments, which addressed his or her previous concerns.